# Efficient Multi-Scale Deformable Attention on GPUs

## Abstract

Multi-scale deformable attention (MSDA) is a core operator in DETR-family vision transformers whose scattered bilinear sampling pattern defeats the tile-based strategies on which FlashAttention-style kernels depend. We present a diagnostic study of GPU kernel optimization for MSDA on NVIDIA A100 (SM 8.0) and H100 (SM 9.0), identifying two failure modes of conventional heuristics and a root cause that is both hardware- and compiler-gated. Dispatch-order reordering does not pay. Across all seven query orders (linear, Morton Z-order, scanline, Hilbert, centroid, a random permutation, and a clustering-and-packing analogue of prior locality methods), sampling-point counts of 4, 8, and 16, and level counts of 2 through 5, no order is statistically distinguishable from linear at the decoder operating point. L2 locality is tile-set by the query-block kernel, not by the dispatch order. Throughput proxies mislead. An 85 %-occupancy point-parallel tiling delivers only 5.1 % of A100 peak bandwidth, while a 17 %-occupancy query-block tiling delivers 36 % and runs 7.4 times faster. The backward-pass bottleneck is scattered-gradient atomic contention. At BF16, the backward kernel attains 2.5 % of A100 peak bandwidth versus 21.3 % on H100. The gap is hardware- and compiler-gated. Ampere has no native BF16 atomic instruction (forcing a 32-bit compare-and-swap emulation), and on H100 the standard CUDA atomic still lowers to that emulation while a relaxed-ordering variant reaches Hopper's native reduction primitive. An FP32-accumulator variant closes the A100 gap entirely. The resulting backends deliver a median BF16 forward speedup of 2.4–2.7-fold at decoder scale and 12–14-fold at encoder scale, and the native-BF16 path cuts encoder-scale peak VRAM by up to 88 % relative to the reference implementation while matching its outputs to floating-point tolerance.

## 1 Introduction

Multi-scale deformable attention (MSDA) is a core component of modern DETR-family detectors and segmentation models, and an efficiency bottleneck: Zhao et al. (2024) report that the MSDA encoder contributes 49 % of GFLOPs but only 11 % of average precision in Deformable DETR, and Cavagnero et al. (2024) identify Mask2Former's deformable-attention pixel decoder as an efficiency bottleneck that limits real-world deployment. Unlike dense self-attention, where FlashAttention-style tiling approaches peak throughput by exploiting contiguous query/key/value layout, MSDA performs scattered bilinear sampling at data-dependent offsets across multiple feature scales; sampled positions are unknown until runtime, precluding the regular tile structure on which efficient attention kernels depend.

We present a diagnostic empirical study of GPU kernel optimization for this operator class on NVIDIA A100 (SM 8.0) and H100 (SM 9.0), isolating two failure modes of conventional optimization heuristics and a hardware-and-compiler-gated root cause of the backward bottleneck. Dispatch-order reordering does not pay, because L2 locality is set by the query-block tiling rather than by the global dispatch order (Section 3.1); high occupancy is not high throughput for this scattered-access operator (Section 3.2); and the backward bottleneck traces to the half-precision atomic-addition path (Sections 3.3 and 4.2), gated both by the hardware (Ampere lacks a native BF16 atomic instruction, forcing a compare-and-swap retry loop) and by the compiler (under CUDA 12.8, only a relaxed-ordering atomic addition reaches Hopper's single-instruction L2 reduction). An FP32-accumulator variant closes the A100 gap, making the backward choice an accumulator-precision decision rather than a programming-model preference. Applied together, our kernel techniques deliver a

median 2.4–2.7× forward speedup at decoder scale and 12–14× at encoder scale, with up to 88 % peak VRAM reduction.

The two NVIDIA generations are a controlled comparison rather than a generality sample. A100 (SM 8.0) has no native BF16 atomic instruction while H100 (SM 9.0) does, so contrasting the same code on the two isolates the hardware gating of the backward bottleneck rather than sampling across vendors (AMD and Intel profiling remains future work, Section 7).

**Contributions.** This study is diagnostic. Each finding isolates one conventional GPU-optimization heuristic and measures, by controlled comparison, whether it survives MSDA's scattered memory access. The three diagnostic findings are as follows.

- **Dispatch-order reordering does not pay.** Across seven query-dispatch orders (linear, scanline, Morton Z-order, Hilbert, centroid, uniform-random, and a DANMP-style K-means clustering-and-packing analogue), sampling-point counts of 4, 8, and 16, and level counts of 2 through 5, no order is statistically distinguishable from linear at the decoder operating point. The query-block kernel, not the global dispatch order, sets L2 locality (Section 3.1 and Table 11).

- **High occupancy is not high throughput.** An 85 %-occupancy tiling reaches only 5.1 % of A100 peak bandwidth, while a 17 %-occupancy tiling runs 7.4× faster. Occupancy-style throughput proxies mislead for this operator (Section 3.2 and Table 1).

- **The BF16 backward is hardware-gated but fixable.** It reaches only 2.5 % of A100 peak bandwidth versus 21.3 % on H100, because Ampere emulates half-precision atomics in software while Hopper provides them natively, a root cause invisible to FLOP-based metrics (Sections 3.3 and 4.2). An FP32 accumulator restores it, lifting the Deformable-DETR backward step on A100 from 0.10× to 0.88× of the reference (Section 3.3 and Table 8).

Beyond these findings, an accompanying open-source library provides CUDA and Triton kernels with up to 14× forward speedup and 88 % lower peak VRAM relative to the reference (Section 5). Swapped into detector training, they cut per-epoch wallclock by up to 20 % (Tables 8 and 18).

## 2 Background

### 2.1 Multi-Scale Deformable Attention

Multi-Scale Deformable Attention (Zhu et al., 2021) replaces the $\mathcal{O}(N^2)$ dot-product attention of the original DETR (Carion et al., 2020) with a fixed number of bilinear lookups across multiple feature scales. Given $N_q$ query elements and $L$ feature maps $\{\mathbf{x}^l\}_{l=1}^L$ at different resolutions, the operator is

$$\mathrm{MSDA}\big(\mathbf{z}_q,\, \hat{\mathbf{p}}_q,\, \{\mathbf{x}^l\}_{l=1}^L\big) = \sum_{m=1}^{M} \mathbf{W}_m \left[ \sum_{l=1}^{L} \sum_{k=1}^{K} A_{mlqk}\, \mathbf{W}'_m\, \mathbf{x}^l\big(\phi_l(\hat{\mathbf{p}}_q) + \Delta\mathbf{p}_{mlqk}\big) \right], \tag{1}$$

where $m \in \{1, \ldots, M\}$ indexes attention heads and $k \in \{1, \ldots, K\}$ indexes sampling points per head per level; $\hat{\mathbf{p}}_q \in [0,1]^2$ is a normalized 2D reference point for query $q$; $\Delta\mathbf{p}_{mlqk} \in \mathbb{R}^2$ are 2D offsets predicted by a linear projection of $\mathbf{z}_q$; $A_{mlqk} \geq 0$ with $\sum_{l=1}^L \sum_{k=1}^K A_{mlqk} = 1$ are scalar attention weights, also predicted from $\mathbf{z}_q$; $\phi_l(\hat{\mathbf{p}}_q)$ rescales the reference point to level $l$'s pixel coordinates; and $\mathbf{x}^l(\mathbf{p})$ denotes bilinear interpolation on $\mathbf{x}^l$ at a 2D coordinate $\mathbf{p} \in \mathbb{R}^2$ (reading the four nearest integer-coordinate neighbors and computing a weighted average). In typical Deformable DETR configurations, $M{=}8$, $L{=}4$, $K{=}4$, giving 128 bilinear lookups per query element. Compared with dense multi-head self-attention (which scales as $O(MN_q^2)$ in time and memory), MSDA requires only $O(MN_qLK)$ work. The cost is qualitative: the $LK$ sampled locations are determined by learned offsets that vary per query, producing *scattered* irregular memory accesses rather than the regular, tile-friendly access pattern of matrix multiplication (Figure 1).

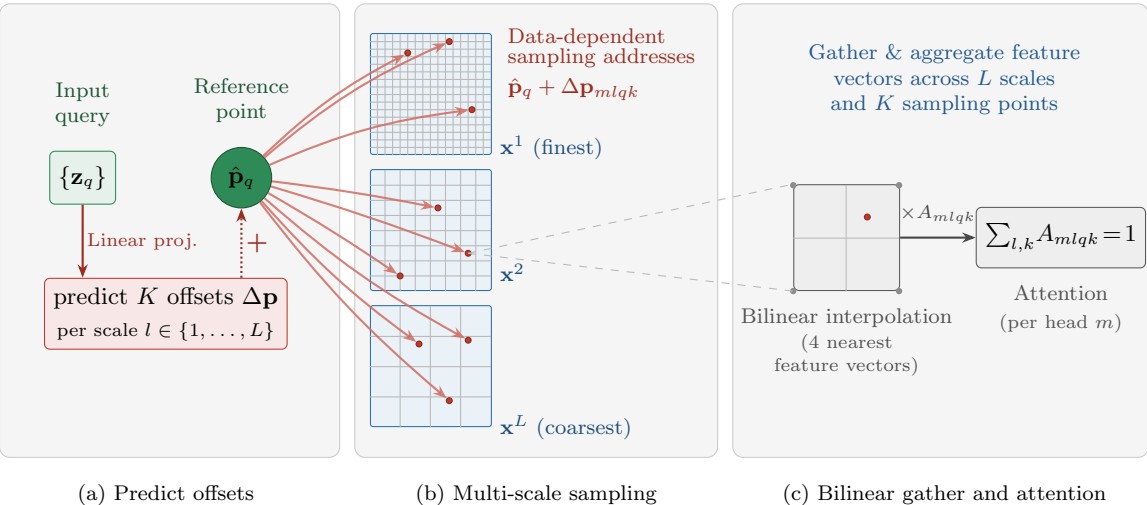

(a) Predict offsets    (b) Multi-scale sampling    (c) Bilinear gather and attention

Figure 1: Multi-scale deformable attention. From query $\mathbf{z}_q$ and its normalized reference point $\hat{\mathbf{p}}_q$, a linear projection predicts $K$ sampling offsets $\Delta\mathbf{p}$ per level (a) across $L$ feature maps of different resolutions (b). Each sampled location is fractional, so bilinear interpolation reads it from the four nearest grid neighbors (c) and weights the four values by $A_{mlqk}$ (summing to one per head $m$). The sampled addresses are data-dependent and scattered across scales, defeating the contiguous-tile structure of FlashAttention-style kernels.

**Computational profile.** The forward pass is dominated by irregular DRAM reads: for each of the $N_q$ queries, the kernel reads $M \cdot L \cdot K \cdot 4 = 512$ feature map values (four neighbors per sample point) from potentially non-contiguous addresses. The backward pass additionally requires scattered atomic writes: gradients for each sampled feature map position must be atomically accumulated into the output gradient tensors, causing contention when nearby queries sample overlapping regions. Neither phase contains the large contiguous tile accesses that enable FlashAttention-style SRAM tiling (Dao, 2024; Shah et al., 2024). The optimization challenge is therefore fundamentally different from dense attention: progress requires reducing per-access cost and serialization overhead, not tiling the computation.

## 2.2 GPU Programming Models

We compare two programming models throughout the paper. *CUDA C++* exposes the thread/warp/block hierarchy and gives the programmer explicit control over shared memory, warp-level shuffle primitives, and atomic operations (atomic-add and compare-and-swap). For MSDA this control is necessary to broadcast sampling coordinates within a warp and to select per-SM atomic primitives, at the cost of hardware-specific engineering that must be revisited each GPU generation. *Triton* (Tillet et al., 2019) operates on 1-D blocks; its compiler maps blocks to warps, coalesces block-internal memory accesses, and auto-tunes block sizes, excelling at regular-tile workloads. Triton also exposes atomic operations with a configurable memory-ordering qualifier that controls the coherence fences emitted around each transaction. For MSDA, Triton's tiling advantage applies naturally to the *query* dimension (consecutive queries are contiguous), but the sampled feature-map addresses remain data-dependent in both models. The two programming models thus face the same data-dependent I/O problem and differ only in how directly each exposes the scheduling primitives that move bytes.

## 2.3 Access-Pattern Characterization

We characterize the reference MSDA implementation from Zhu et al. (2021) on an NVIDIA A100-SXM4-40GB (SM 8.0, 1.56 TB/s peak memory bandwidth) using hardware-counter profiling at the standard decoder configuration ($B$=4, $N_q$=300, $H$=$W$=800, $d$=256, $L$=4, BF16). The reference forward kernel sustains only 29.3 % of peak memory bandwidth (Figure 5) while reaching well below the A100 compute ridge, and the

reference backward kernel is twice as expensive as the forward for structural reasons we preview here and quantify in Section 3. Two aspects of the access pattern are load-bearing for the rest of the paper and together motivate the diagnostic structure of Section 3.

**Read-side: scattered loads with no stable reuse.**  For each of the $N_q$ queries, the forward pass issues $M \cdot L \cdot K \cdot 4 = 512$ bilinear-neighbor reads at addresses that are the output of a learned offset network. Two queries may sample nearby image regions on one iteration and disjoint regions on the next, so there is no iteration-stable reuse pattern for cache or prefetcher policies to exploit. The only regular structure is the sequential advance of the *query index*, which the L2 pre-fetcher can follow as long as the kernel dispatches queries in their original order. Any reordering that attempts to manufacture spatial locality (Section 3.1) replaces this monotonic stream with a data-dependent permutation and cannot recover the lost streaming behavior with software cache hints.

**Write-side: scattered atomic gradient accumulation.**  The backward pass inverts the access direction: each contributing bilinear neighbor of every sampled position receives an atomic accumulation of its gradient share. Nearby queries that happen to sample overlapping feature regions contend on the same cache lines, and because sampling locations are data-dependent the set of contenders is not knowable at dispatch time. Under the encoder-scale feature maps that dominate training wall-clock time, dozens of warps can contend on the same atomic target within microseconds. The cost of this contention depends entirely on the hardware's atomic primitive for the accumulation data type (Section 3.3), not on the programming model used to write the kernel.

**A secondary buffer observation.**  In addition to the two access-pattern properties above, the reference implementation concatenates sampling coordinates and attention weights into a single $(B, N_q, L, K, 3)$ tensor (the three components are the two spatial offsets $(x, y)$ and the scalar attention weight) before launching the CUDA kernel. The buffer is unnecessary (a kernel can read coordinates and weights from disjoint pointers at identical cost), and eliminating it removes a concatenation allocation from the forward critical path. We include this observation because it informs the kernel design described in Section 2.4, but it is not a central diagnostic finding of the paper: its performance impact is within a few percent of peak at the configurations we measured.

### 2.4   Kernel Design

The kernels we benchmark apply three standard optimizations to the access pattern above: vectorized 128-bit feature-map loads, warp-level sharing of the $M \cdot L \cdot K$ sampling parameters, and relaxed-ordering atomic addition in the gradient-scatter loop (safe because gradient accumulation is commutative and the kernel-end barrier carries the happens-before ordering). The CUDA backward additionally upcasts gradient tensors to FP32 before the kernel launch, routing through the hardware-native FP32 atomic-addition primitive and down-converting on return.

These techniques narrow but do not eliminate the BF16 atomic cost on SM 8.0; the residual gap is hardware-gated (Section 3.3). Full pseudocode for the two patterns appears in Section E; Section D isolates each technique's contribution.

## 3   Failure Modes and Mechanisms

This section presents two failure modes that we quantify while optimizing MSDA for commodity NVIDIA GPUs, and the hardware-gated root cause of the second. The failures themselves are, in hindsight, intuitive: scattered access has no exploitable locality, and occupancy need not predict bandwidth. The literature has not previously measured either effect for an attention operator. Each subsection is organized around the same three questions: what fails, by how much, and why. The failure modes are mechanistically distinct (one reflects a property of the operator, the other a property of the hardware), and separating them is the central diagnostic contribution of this paper.

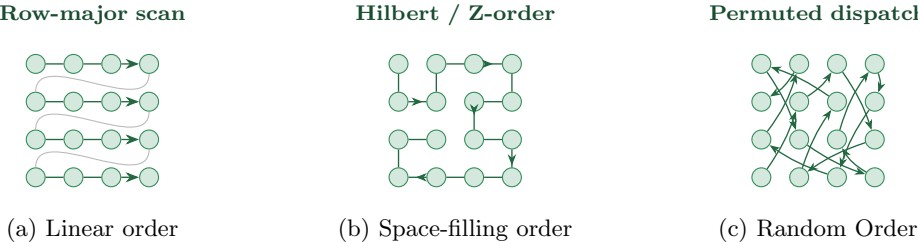

| Row-major scan | Hilbert / Z-order | Permuted dispatch |
| --- | --- | --- |
| (a) Linear order | (b) Space-filling order | (c) Random Order |

Figure 2: Three of the seven query-dispatch orders over the same set of queries: (a) a linear/scanline traversal, (b) a locality-preserving space-filling traversal (Hilbert / Morton Z-order), and (c) a random permutation. Only the global query-to-thread-block assignment changes. The query-block kernel and its tiling stay fixed, and the sampled feature-map addresses remain data-dependent. Because the tiling, not the dispatch order, sets L2 locality, all seven orders are statistically indistinguishable from the linear baseline at the decoder operating point.

## 3.1 Locality failure: dispatch-order reordering does not pay

The first optimization we attempted is among the most widely recommended for memory-bound kernels: reorder the work items so that neighboring threads access neighboring addresses. For MSDA this is especially tempting because the dominant cost is scattered bilinear reads of multi-scale feature maps. If queries whose sampling locations fall near one another in image space were grouped onto the same thread blocks, the argument goes, their neighbor fetches would overlap in L2 and bandwidth demand would fall.

**Measurement.** We tested seven query-dispatch orders on the same Triton query-block forward kernel at $K=4$, $L=4$, default-configuration encoder/decoder. Writing $\bar{\mathbf{p}}_q = (\bar{x}_q, \bar{y}_q)$ for query $q$'s mean sampling position and $\bar{\mathbf{r}}_q$ for its mean cross-level reference point, each order dispatches the $Q$ queries in the order $\text{argsort}_q k(q)$ for a sort key $k$:

$$
\begin{aligned}
k_{\text{lin}}(q) &= q, & k_{\text{scan}}(q) &= \lfloor \bar{y}_q \rfloor W + \lfloor \bar{x}_q \rfloor, \\
k_{\text{mort}}(q) &= \text{Z}(\bar{\mathbf{p}}_q), & k_{\text{hilb}}(q) &= \text{H}(\bar{\mathbf{p}}_q), \\
k_{\text{cent}}(q) &= \text{cluster}(\bar{\mathbf{r}}_q), & k_{\text{cap}}(q) &= \text{KMeans}_{k'}(\bar{\mathbf{r}}_q), \ k' \approx 2\sqrt{Q},
\end{aligned}
\tag{2}
$$

where $W$ is the feature-map width, Z and H are the Morton (Z-order) and Hilbert space-filling indices, and cluster and $\text{KMeans}_{k'}$ assign reference-point cluster labels so the centroid and CAP orders dispatch cluster-by-cluster (CAP is the GPU analogue of DANMP's §6.3 packing scheme (Li et al., 2026)). The seventh order is a uniform random permutation, a lower bound on any structured ordering. Figure 2 contrasts three of them over a shared query set. Across both GPUs, all three precisions, and every tested decoder and encoder configuration ($B \in \{4, 8\}$ decoder, $B=2$ encoder), the seven orders are statistically indistinguishable from linear. The median deviation is below 1%, and no order, geometric or clustering, is systematically faster or slower than the baseline. A handful of cells exceed $\pm 2\%$ (at most $\approx 3\%$), but these are not order-specific. They land on geometric and clustering orders alike and in both directions, and they do not reproduce across the $K \in \{4, 8, 16\}$ sweeps, the signature of run-to-run measurement variation rather than dispatch-order latency. Per-cell ratios, $p_{99}$ latencies, 95% confidence intervals on each ratio, and a contamination flag for the high-$p_{99}$ cells appear in Table 11 of Section A.

The null is not specific to $K=4$. Extending the sweep to $K=8$, $K=16$, and $L \in \{2, 3, 5\}$ leaves every standard-scale and batch$\geq 2$ cell within $\pm 2\%$ across both GPUs, apart from one flagged jitter outlier starred in the $L$-sweep table (Tables 12 to 14 of Section A). At $K=16$ the worst-case deviation is 1.6% (H100, FP32, Morton). At $L=3$ (Mask2Former) it is 1.1% and at $L=5$ it is 1.3% (decoder-scale). The $K$-sweep encoder-scale configuration ($B=2$, 1536×2048, $L=4$) also stays within $\pm 2\%$ at $K=4$, 8, and 16. At $L=5$ and the full encoder resolution (1536×2048, whose five-level pyramid reaches $N \approx 262k$), cache-friendly orderings are measurably faster than linear (7–10% on both GPUs). This is a distinct operating-point effect, not the dispatch-order invariance claim, and Section 5 discusses it. No query-dispatch permutation tested, including

the random permutation that serves as a lower bound on any structured ordering, produces a systematic change in forward-pass latency at the decoder operating points where the invariance claim applies.[1]

**Mechanism.**   The null result has a structural explanation. The sampling locations of MSDA are the output of a learned offset network, and the reference points of each query are already monotonic in the query index (Section 2). From the memory subsystem's perspective, the sampled feature-map addresses are data-dependent and effectively random regardless of query ordering; no static permutation of the dispatch order can manufacture cache-line sharing that the kernel's tile-set does not already provide, because L2 locality on the MSDA forward kernel is set by the query-block tiling (Section 3.2), not by the global sort order of queries into blocks. The hardware L2 pre-fetcher exploits the monotonic advance of the *query index* itself (contiguous loads over sampling parameters and output buffers), and this streaming pattern is equally available under any permutation because each query's parameter block is fetched independently. The random-permutation control confirms this: a permutation that destroys all spatial structure performs identically to linear dispatch, proving that the kernel's memory traffic is query-order-invariant. Li et al. (2026) reach a compatible conclusion on a near-memory- processing substrate: MSDA's data-dependent targets defeat locality-based reuse under conventional caching.

**Scope.**   The null scopes to *dispatch-order reordering* (permuting the global query-to-thread-block assignment, kernel held fixed); it is not a negative result for memory-access reorganization in general. The CAP analogue tested here clusters reference centroids with fixed-$k$ K-means and dispatches queries sorted by cluster label, deviating on four axes from DANMP's full sampling-point + DBSCAN + 20%-subsample + packing recipe. A partial port that closes three of the four deviations (Section F) leaves the null intact under realistic input-dependent offsets, so the §3.1 result does not falsify the broader claim that memory-access reorganization via a packing kernel or channel-dimension remap (Xiong et al., 2024; Huang et al., 2025) could benefit structurally-related operators.

**Implication.**   Practitioners should not invest in space-filling-curve or cluster-label-sorted dispatch strategies for MSDA's forward pass at the validated $K \in \{4, 8, 16\}$, $L \in \{2, 3, 4, 5\}$ configurations. The irregular access pattern is tile-set rather than order-set, and no query-dispatch permutation we tested, including a faithful GPU port of DANMP's clustering substep, unlocks speedup. Memory-access reorganization through a different primitive — a packing kernel that physically fuses bilinear reads sharing sub-targets, or the channel-dimension remap that DCNv4 exploits on deformable convolution — remains open for future work.

## 3.2   Throughput-proxy failure: high occupancy is not high throughput

The second failure concerns the metrics by which GPU kernels are usually tuned. SM occupancy (the fraction of the theoretical maximum resident warps per SM that a kernel actually sustains) is the single most widely cited kernel health indicator, and the standard optimization playbook recommends raising it until it saturates. For MSDA on the decoder configuration, Figure 3 contrasts two Triton tilings that span a wide occupancy range. In a *point-parallel* tiling each program instance processes a single (query, head, point) triple; in a *query-block* tiling each program instance fuses all points and heads for a contiguous block of queries.

The point-parallel tiling achieves a sustained SM occupancy of 85.2 % on the A100; the query-block tiling achieves only 17.1 %. By the occupancy heuristic the point-parallel tiling is preferable; in practice it is the point-parallel tiling that is a *throughput failure*.

---

[1]Per-cell relative standard deviation ($\sigma/\mu$) is in the 1–3% band for the large majority of cells on the shared compute nodes (GPU clocks are not externally locked, so the measured dispersion exceeds nvbench's 0.5% reproducibility floor). The $n{=}100$-trial median is nevertheless discriminating to a much tighter level. Median linear-baseline $\sigma$ is 0.0113 ms on A100 and 0.0073 ms on H100; with $n{=}100$ and a 95% two-sided confidence interval, the minimum detectable effect on the mean $p_{50}$ is therefore $\approx 0.41\%$ on A100 and $\approx 0.34\%$ on H100, well below the $\pm 2\%$ band. This bound characterizes the well-behaved majority. It does *not* license the null in the minority of FP16 cells whose $\sigma/\mu$ reaches 45–55%, where a 2% effect cannot be excluded at 95%. Those cells carry a $p_{99}$ tail-spike signature of run-to-run measurement variation (Table 11) rather than order-dependent latency. The analysis flags and excludes them from the categorical claim rather than counting them toward it. Table 11 tabulates per-cell 95% confidence intervals on each ratio (propagated from the reported $\sigma$ under a normal approximation) alongside $p_{99}$.

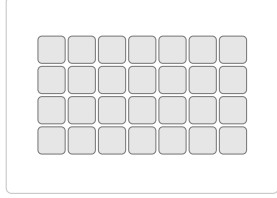
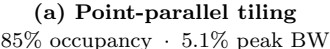
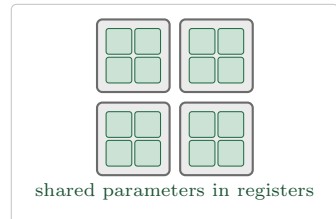

**(a) Point-parallel tiling**
85% occupancy · 5.1% peak BW

**(b) Query-block tiling**
17% occupancy · 36% peak BW

Figure 3: Two Triton tilings of the MSDA forward pass. **(a)** A *point-parallel* tiling spreads single sample points across many small programs, reaching 85 % SM occupancy but only 5.1 % of A100 peak bandwidth. **(b)** A *query-block* tiling gives each program a block of queries and splits the channel dimension across a warp's lanes, reusing sampling parameters in registers. It sustains only 17 % occupancy yet reaches 36 % of peak bandwidth and runs 7.4× faster. Occupancy is therefore a misleading proxy for throughput on this memory-bound operator.

Table 1: Roofline metrics for two Triton forward tilings at decoder scale, BF16, on both GPUs. "Block" denotes the query-block tiling and "Point" the point-parallel tiling; both are defined at the start of Section 3.2. Peak memory bandwidth is 1.56 TB/s (A100) and 3.35 TB/s (H100). The kernel with lower occupancy delivers higher bandwidth and lower latency.

| GPU | Tiling | Occupancy (%) | L2 hit (%) | BW (% peak) | Time (ms) |
|---|---|---|---|---|---|
| A100 | Point | 85.2 | 87.6 | 5.1 | 0.613 |
| A100 | Block | 17.1 | 34.1 | 36.1 | 0.083 |
| H100 | Point | 83.3 | 93.2 | 3.8 | 0.356 |
| H100 | Block | 14.2 | 35.1 | 18.7 | 0.069 |

**Measurement.**  Table 1 summarizes the roofline metrics for both tilings on the A100 at BF16, decoder configuration. The point-parallel tiling sustains 85 % occupancy and 87 % L2 hit rate, but delivers 78.9 GB/s of DRAM bandwidth, 5.1 % of the A100 peak (1.56 TB/s). The query-block tiling runs at 17 % occupancy, 34 % L2 hit rate, and delivers 561.7 GB/s, or 36.1 % of peak, a 7.1× improvement in effective bandwidth at one-fifth of the occupancy. Forward latency follows bandwidth rather than occupancy: the point-parallel tiling takes 0.613 ms, the query-block tiling 0.083 ms, a 7.4× speedup for the kernel that looks less healthy on the occupancy metric. The H100 roofline is qualitatively identical (Table 1).

**Mechanism.**  The point-parallel tiling satisfies every condition the occupancy heuristic is designed to detect (warps resident, scheduler slots filled, high L2 hit rate because each thread touches four bilinear neighbors) but issues those loads as tiny independent transactions, one per program, and the memory pipeline fills with many small in-flight requests that cannot be coalesced post hoc. The query-block tiling fetches sampling parameters once per block, broadcasts coordinates across the block's threads, and issues coalesced wide bilinear loads; resident warp count drops because the per-program register footprint is larger, but each request moves more useful bytes. Hardware-counter warp-stall attribution confirms the mechanism directly: 67.2 % of active warp-issue slots on the point-parallel A100 kernel stall on *long-scoreboard* dependencies, the stall class that indicates a warp waiting for an outstanding DRAM load. High occupancy does not hide DRAM latency if every warp is waiting for a different cache miss.

**Implication.**  For memory-bound operators with irregular access patterns, the binding constraint is the DRAM transaction size distribution, not warp availability, and the optimization that raises occupancy may lower bandwidth. Although the low-occupancy-beats-high-occupancy principle is well established for general GPU kernels (Volkov, 2010; Shobaki et al., 2020) and for dense attention (Spector et al., 2025), ours is the first quantitative demonstration of the divergence for a scattered-access attention operator. It is the reason we settled on query-block tiling for the forward pass even though no conventional tuning metric preferred it.

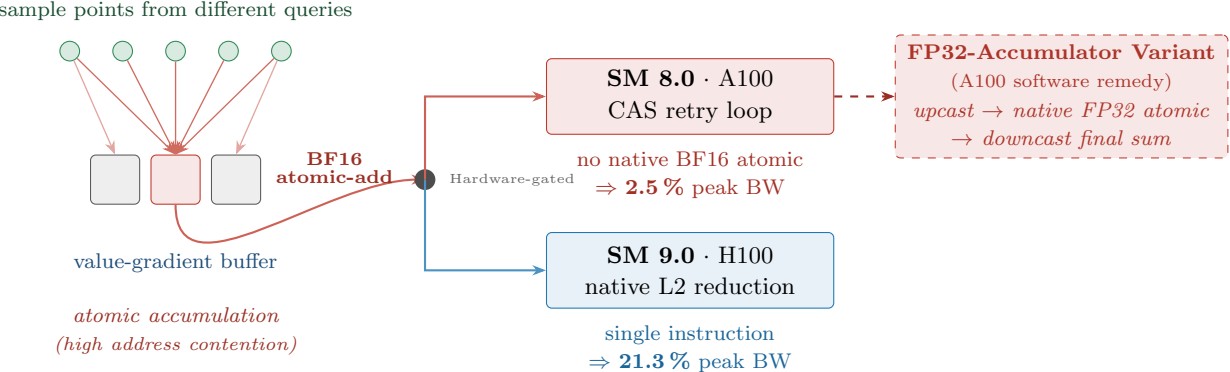

Figure 4: Backward-pass atomic accumulation. Because sampling locations are data-dependent, gradients from many queries converge on the same feature-map positions and must be accumulated atomically, serializing under contention. The cost of a BF16 atomic add is hardware- and compiler-gated. SM 8.0 (A100) has no native BF16 atomic, so it emulates the add as a compare-and-swap retry loop and the backward kernel reaches only 2.5 % of peak bandwidth, whereas SM 9.0 (H100) recovers 21.3 % through a native L2 reduction primitive. An FP32-accumulator variant routes through the hardware-native FP32 atomic and closes the A100 gap.

### 3.3  Atomic contention: hardware-gated under native precision

The forward-pass throughput-proxy failure has a backward-pass counterpart that is larger in magnitude, more consequential for training, and caused by a different mechanism. The MSDA backward pass accumulates gradients at the sampled feature positions using atomic adds: every query that samples a given neighbor contributes additively to that neighbor's gradient, and because sampling locations are data-dependent the set of contending queries is not known until runtime. Under dense workloads (encoder-scale $1536 \times 2048$ feature maps with hundreds of queries per point), the same cache line is updated by many warps within microseconds of each other. The performance of these atomics depends on hardware support. Before SM 9.0 (Hopper), the GPU had no native BF16 atomic-add instruction; the CUDA runtime emulated it with a multi-step compare-and-swap (CAS) loop that retries on every concurrent modification (Section 4.2 confirms this empirically: on Hopper the compiled kernel reduces to a single hardware atomic; on Ampere the same source compiles to an approximately ten-instruction CAS retry loop). Figure 4 sketches the contention and the two lowering paths.

SM 9.0 adds a single-transaction hardware primitive for BF16 atomics. Separately, Triton exposes a relaxed memory-ordering qualifier on its atomic-addition intrinsic, weakening the default acquire-release semantics to permit the memory subsystem to coalesce atomics more aggressively.

**Measurement.** Our Triton backward kernel at decoder scale on the A100 sustains 401 GB/s (25.8 % of peak) at FP32 and 38.5 GB/s (2.5 % of peak) at BF16, a drop of $\sim$10$\times$ in effective bandwidth purely from switching the accumulation precision from FP32 to BF16 on the same kernel, same GPU, and same configuration (Table 2). At encoder scale the collapse translates directly into wall-clock time. The same Triton backward kernel takes 123.4 ms to process one encoder batch on the A100 at BF16. The CUDA backward kernel sidesteps the BF16 atomic contention regime entirely by upcasting gradient tensors to FP32 in the Python wrapper before launching the kernel, and takes 17.2 ms on the same input — a 7$\times$ gap. On the H100 the same comparison inverts entirely: the Triton kernel completes in 4.33 ms and the CUDA kernel in 10.19 ms (2.4$\times$ the other direction). Neither kernel was modified between GPUs; the code path is bit-identical.

**Mechanism.** The root cause is a single hardware capability that A100 (SM 8.0) lacks and H100 (SM 9.0) supplies: a native BF16 atomic-add instruction. On SM 8.0 the runtime emulates a BF16 atomic-add as a compare-and-swap (CAS) loop (NVIDIA Corporation, 2025a); under MSDA's scattered gradient pattern,

Table 2: Roofline metrics for the Triton backward kernel at decoder scale ($B{=}4$, 800×1333, BF16/FP32). At FP32 both GPUs sustain 26 % to 29 % of peak bandwidth. At BF16 the A100 collapses to 2.5 % while the H100 holds at 21.3 %, an order-of-magnitude gap at matched code.

| GPU | Precision | BW (GB/s) | BW (% peak) | Time (ms) |
|---|---|---|---|---|
| A100 | FP32 | 401.4 | 25.8 | 0.791 |
| A100 | BF16 | 38.5 | 2.5 | 3.997 |
| H100 | FP32 | 984.4 | 29.4 | 0.292 |
| H100 | BF16 | 714.2 | 21.3 | 0.190 |

Table 3: Measured L2 atomic-add / hardware-reduction sectors for the MSDA backward kernel (Triton query-block, decoder scale), collected under NSight Compute. Counts in millions. The emulated compare-and-swap (CAS) retry loop fires on *only* the A100 BF16 path (Section 3.3).

| GPU | Prec. | CAS (M) | Red. (M) | Mechanism |
|---|---|---|---|---|
| A100 (SM 8.0) | BF16 | 34.8 | 0 | CAS retry loop |
| | FP32 | 0 | 34.1 | native reduction |
| H100 (SM 9.0) | BF16 | 0 | 4.3 | native reduction |
| | FP32 | 0 | 8.5 | native reduction |

where dozens of warps write to the same cache line in a small time window, retry serialization dominates. Counter instrumentation confirms this directly (Table 3): at the A100 decoder BF16 operating point the backward kernel issues 34.8M L2 compare-and-swap atomic sectors and essentially zero hardware-reduction sectors, whereas the identical scatter routes through 4.3–34.1M native *reduction* sectors, with zero CAS, under FP32 on A100 and under both precisions on H100 (the native counts scale with operand width and Hopper's more aggressive atomic coalescing). The SM 8.0 BF16 path is the only configuration that emulates the accumulation, converting the contention claim from an inference about the bandwidth collapse into a measured effect. The CUDA backward kernel sidesteps the loop by upcasting inputs and the upstream gradient to FP32 in the host-side wrapper, then issuing hardware-native FP32 atomics and down-converting on return. A control backend that disables the upcast and forces native-BF16 atomic-add lands at 1.706 ms / 94 GB/s on A100 decoder BF16, splitting the 5.3× CUDA/Triton decoder gap roughly evenly between the FP32-upcast precision rescue and the nvcc-vs-Triton codegen advantage within BF16 atomics. SM 9.0 promotes the BF16 atomic to a native single-transaction hardware primitive (NVIDIA Corporation, 2025b; 2022) that completes regardless of contention, and the gap closes (Section 4.2 refines the mechanism: the gap is hardware-*and*-compiler-gated, requiring Triton's specific atomic lowering path).

**Software mitigations.** The bandwidth collapse is gated only when backward atomics operate at native BF16 precision; accumulating gradients into an FP32 scratch buffer (the *FP32-accumulator* variant) replaces the contended BF16 CAS loop with a hardware-native FP32 atomic-add at the cost of a larger buffer. At encoder scale on A100 it runs 18.2 ms, 6.8× faster than the native-precision Triton backward (123.4 ms) and within 6 % of our CUDA backward (17.2 ms) at 45 % of its peak workspace; at decoder scale ($B{=}8$, BF16) it runs 1.54 ms, 33 % faster than native Triton and 3 % faster than CUDA. On H100, where hardware BF16 atomics are already fast, the path regresses 18 % at encoder scale (5.1 ms vs 4.3 ms). The FP32 buffer is transparent to PyTorch's automatic-mixed-precision context and accumulates gradients in full precision (so, unlike the native-BF16 path, it is order-independent) while matching the FP32 reference to tolerance on captured real inputs (Table 9). It raises encoder backward peak VRAM from the native-BF16 path's 448 MB to 576 MB (+29%), still 45 % below the CUDA workspace. Three reduce-by-key alternatives (warp, block, and sort/segscan, all numerically matched to the baseline) fail: on A100 decoder BF16 they run 2.8–12.9× slower than the plain atomic (5.56 ms/22.59 ms/4.85 ms versus 1.75 ms); on H100 1.5–6.1× slower. The software-dedup structures cost more than the contention they eliminate even on SM 8.0. The only software lever that pays at MSDA's contention regime is accumulator precision, not pre-reduction.

**Implication.** The three failure modes are mechanistically distinct. Locality is a property of the operator's access pattern; throughput-proxy is a property of the tuning metric; atomic contention is a joint property of target architecture and accumulator precision. Under native-precision BF16 atomics the backward gap is an order of magnitude and resolved only on SM 9.0; under FP32 accumulation it closes on both GPU generations. The practical consequence is that the backward-pass choice is an *accumulator-precision* decision gated by the target hardware: use FP32 accumulators on A100 and native BF16 atomics on H100; Triton is the fastest or tied backend in both cases (Section 4.1). Both paths use relaxed memory ordering, which is safe because gradient accumulation is commutative and the kernel-end barrier provides the required happens-before relationship (NVIDIA Corporation, 2025a). No prior attention-kernel study isolates this effect, and it independently validates concurrent atomic-reduction work for differentiable rendering (Durvasula et al., 2025) on a second operator class.

## 4 Implications and Generalization

### 4.1 Backward-pass programming model: an accumulator-precision question

Which backend should a downstream user pick on which GPU? Section 3.3 established that the backward-pass atomic contention is hardware-gated under native-precision atomics, and that an FP32-accumulator variant closes the gap on A100. This subsection turns those findings into wall-clock guidance for the two programming models we implemented.

**Forward pass.** On both A100 and H100, the Triton forward kernel using query-block tiling matches or slightly beats the best CUDA forward kernel at the standard decoder configuration. At $B=4$, BF16, decoder scale the Triton forward latency is 0.534 ms on A100 versus 0.564 ms for the corresponding CUDA kernel, and 0.445 ms versus 0.473 ms on H100 (5 %–6 % Triton advantage on both GPUs). The advantage comes from the Triton compiler's better exploitation of the query-block tiling described in Section 3.2: the same technique is available in CUDA but requires hand-written warp-broadcast code that is harder to tune. For the forward pass the programming-model choice is therefore a minor latency difference; both models have access to the same bandwidth, and both deliver it.

**Backward pass, decoder scale.** At small batch sizes the backward pass does not saturate the atomic pipeline, and the Triton and CUDA backends are within single-digit percent of each other on both GPUs. At $B=8$ the CUDA backward edges ahead on A100 (1.58 ms vs 2.04 ms), but on the H100 the two kernels are statistically tied. The $\approx 1.5\%$ gap is well within the per-cell run-to-run variation of the backward decoder cells (whose $\sigma$ is several percent) and is not stable across configurations. The decoder backward therefore admits no winner. Consistent with Section 5, where the custom backends agree to within 4–6%, the *encoder* scale, not the decoder backward, decides the architectural ordering.

**Backward pass, encoder scale.** At encoder scale the native-precision backward gap is 7× rather than tens of percent, and, unlike at decoder scale, the two GPUs genuinely disagree on which kernel wins. On the A100 the CUDA backward processes a $B=2$, 1536×2048 encoder batch in 17.2 ms while the native-precision Triton backward takes 123.4 ms; on the H100 the same two kernels run in 10.2 ms and 4.33 ms, respectively. The FP32-accumulator Triton variant (Section 3.3) closes the A100 gap on latency to within 6 % of CUDA (18.2 ms vs 17.2 ms) while using 45 % less peak VRAM (Table 6). Table 4 presents all three backends on both GPUs. Two observations follow. First, under native-precision atomics neither kernel is universally preferable, but the FP32-accumulator variant resolves the crossover in favor of Triton on both GPUs. Second, the native-precision gap is dominated entirely by the atomic-hardware discontinuity (Section 3.3): at FP32, where the SM 8.0 software CAS loop is not in the path, the Triton backward is within a factor of two of the CUDA backward on the A100 even at encoder scale.

### 4.2 Generalization beyond MSDA: the gap is compiler-gated

The backward gap reported in Section 3.3 is hardware-*and*-compiler-gated: Triton's relaxed-semantics atomic reaches Hopper's native BF16 reduction primitive, while nvcc's half-precision atomic-add overload does not.

Table 4: Backward-pass latency at encoder scale ($B$=2, 1536×2048, $L$=4, BF16) for both programming models on both GPUs. Under native-precision atomics, CUDA wins on A100 by 7× and Triton wins on H100 by 2.4×. The FP32-accumulator Triton variant closes the A100 gap, making Triton the fastest or tied backend on both GPUs. Neither kernel was modified between GPUs.

| Backend | A100 bwd (ms) | H100 bwd (ms) |
|---|---|---|
| Reference impl. | 692.3 | 49.0 |
| CUDA path | 17.2 | 10.2 |
| Triton path (BF16 acc) | 123.4 | 4.33 |
| Triton path (FP32 acc) | 18.2 | 5.1 |

The gap should therefore not reproduce on PyTorch-native operators compiled through nvcc. We test this on PyTorch's standard scatter-add operator, the only PyTorch-native scattered reduction whose dispatch chain bottoms out in the same BF16 atomic-add call as our CUDA kernel; the embedding-lookup and index-add backward paths instead go through sort-based reductions with no atomics.

Sweeping the scatter-add operator across a 72-cell matrix of ($M, D, N$) sizes, index distributions (uniform and Zipf-$s$=1.5), and precisions (FP32/FP16/BF16) on both GPUs reveals two properties: *FP32 is indifferent to contention* ($p_{50}$ stays under 2 ms in every cell), and *every half-precision atomic collapses two-to-three orders of magnitude under Zipf on both architectures*, including native FP16 (1499 ms / 1823 ms at the worst-contention cell). The full 72-cell sweep is in Section C, supported by Tables 20 and 21. The A100→H100 BF16 ratio for the scatter-add operator at the worst cell is order-unity (0.9–1.5×), far below the reproducible 7× gap we measure on MSDA's backward at the same precision. The hardware-gated claim, as initially stated, does not reproduce on the PyTorch-native operator.

**Counter-level diagnosis.** Hardware-counter instrumentation at the worst-contention cell locates the divergence. The MSDA backward kernel on H100 emits `red.*`-type sectors (the hardware reduction path), the only quadrant of the table that does; A100 at BF16, and both GPUs for the scatter-add operator, stay on `atom.*` (full counters in Table 21). The mechanism has three components. (i) Hardware: shell-level SASS disassembly of our Triton-compiled H100 kernel shows the atomic gradient writes lower to `REDG.E.ADD.BF16x8.RN.STRONG.GPU`, a single instruction that reduces eight BF16 lanes at once through the L2 cache; Ampere SASS has no equivalent, because SM 8.0 lacks the native BF16 atomic primitive that PTX ISA 7.8 introduces for SM 9.0 and later (NVIDIA Corporation, 2025b). (ii) Triton codegen: the atomic-addition intrinsic in our backward kernel is invoked with relaxed memory-ordering semantics (Triton Project, 2024) (the intrinsic defaults to acquire-release, so the relaxed lowering this paper relies on requires that the explicit qualifier be supplied at the call site). With an unused return, the intrinsic emits an eight-register vectorized PTX atomic-add at BF16 (`atom.global.gpu.relaxed.add.noftz.v8.bf16`) whose return is dead, which the PTX assembler promotes to the hardware reduction form wherever supported. (iii) nvcc codegen: the standard CUDA half-precision atomic-add overload (as shipped with CUDA 12.8) lowers to `ATOM.E.CAS.STRONG.GPU` even when the return value is discarded, empirically confirmed by the zero `red` sector counts for the scatter-add operator. The CUDA-library specialization that selects the native lowering is toolkit-version-dependent. The study reports behavior observed under CUDA 12.8, with later toolkits potentially closing this gap as the standard CUDA half-precision atomic-add overload evolves. Consumers whose compiler marks the atomic as a reducible no-return operation inherit the H100 advantage; consumers that bottom out in the standard half-precision atomic-add overload do not. The practical recommendation therefore stands: on A100, use an FP32-accumulator backward (Section 3.3); on H100, Triton kernels with relaxed-ordering atomics need no workaround, while nvcc-compiled consumers may still benefit until the CUDA runtime adds the `red.bf16` lowering.

## 5 Experimental Evaluation

**Setup.** All measurements run on a single-GPU partition of a shared academic HPC cluster on two nodes: an A100-SXM4-40GB (SM 8.0) and an H100 SXM (SM 9.0), both under CUDA 12.8, PyTorch 2.9.0, and

Table 5: MSDA operator latency ($p_{50}$, ms) and speedup over the reference on the same GPU. Config: B4 800×1333, BF16, eager. Best latency per column in **bold**.

| | A100 | | | | H100 | | | |
| --- | --- | --- | --- | --- | --- | --- | --- | --- |
| | Fwd (ms) | Fwd (×) | Bwd (ms) | Bwd (×) | Fwd (ms) | Fwd (×) | Bwd (ms) | Bwd (×) |
| Reference | 1.416 | 1.00 | 5.440 | 1.00 | 1.053 | 1.00 | 2.142 | 1.00 |
| CUDA (Zhu et al., 2021) | 0.544 | 2.63 | **1.495** | 3.62 | 0.446 | 2.39 | 1.170 | 1.83 |
| CUDA *ours* | 0.564 | 2.53 | 1.522 | 3.56 | 0.472 | 2.26 | 1.206 | 1.78 |
| Triton, point-parallel | 0.561 | 2.55 | 1.777 | 3.07 | **0.441** | 2.42 | **1.160** | 1.84 |
| Triton, query-block | **0.534** | 2.68 | 1.572 | 3.46 | 0.445 | 2.39 | 1.187 | 1.80 |

Table 6: Peak GPU memory (MB) per backend. A100-SXM4-40GB, BF16, B4 800×1333, eager. Best peak per phase in **bold**.

| | Fwd only | | Fwd+Bwd | |
| --- | --- | --- | --- | --- |
| | Alloc (MB) | Peak (MB) | Alloc (MB) | Peak (MB) |
| Reference | 54.7 | 117.9 | 0.6 | **151.7** |
| CUDA (Zhu et al., 2021) | 89.2 | 134.3 | 0.6 | 177.6 |
| CUDA *ours* | 45.0 | **45.9** | 0.6 | 221.7 |
| Triton, point-parallel | 45.0 | **45.9** | 0.6 | **151.7** |
| Triton, query-block | 45.0 | **45.9** | 0.6 | **151.7** |

the matching Triton release. Unless noted, we use eager mode and the standard Deformable DETR decoder configuration: batch size $B=4$, input resolution 800×1333, hidden dimension $D=256$, and $L=4$ feature levels. Latencies are $p_{50}$ over 100 iterations after 10 warmup iterations with explicit device synchronisation barriers; standard deviations and tail statistics are reported in Section A. We compare four implementations: the reference of Zhu et al. (2021), our optimized CUDA kernel, the high-occupancy point-parallel Triton tiling, and our query-block Triton tiling. Full benchmark parameters, the encoder-scale configuration, a cumulative technique ablation, and per-GPU sweep tables are all deferred to Section A. We report absolute latencies alongside every ratio, since the reference backward benefits disproportionately from the A100→H100 upgrade ($\sim 2.5\times$) while our backends benefit by $\sim 1.3\times$ in the same comparison (already closer to the bandwidth ceiling on A100); read as ratios, the H100 speedups therefore appear smaller than the A100 ones even though absolute latencies are lower on H100.

**Operator-level latency (Table 5).** Our Triton query-block tiling is the fastest forward kernel on both GPUs at the decoder scale (0.534 ms on A100, 0.445 ms on H100, corresponding to $2.68\times/2.39\times$ over the reference). At decoder scale the backward-pass latencies of three of the four custom backends (our CUDA kernel, our query-block Triton kernel, and its FP32-accumulator variant) converge to within 6 % of each other on A100 and 4 % on H100 because the atomic pipeline is not saturated; the Triton point-parallel tiling is 13 % to 17 % slower than the custom-backend cluster on A100, consistent with the throughput-proxy failure of Section 3.2. The architectural backward crossover appears only at encoder scale (Table 4). Encoder-scale forward speedup reaches $12.18\times$ on A100 and $14.00\times$ on H100, again driven by the query-block tiling of Section 3.2.

**Memory footprint.** All custom backends cut the decoder forward-only peak from 117.9 MB to 45.9 MB ($-61\%$) by eliminating the sampling-parameter concatenation buffer of Section 2.3. At encoder scale (Table 7) the combined forward + backward peak drops from 3607.8 MB for the reference (FP32 accumulator) to 1055.9 MB for our CUDA path (FP32 accumulator), to 575.9 MB for the Triton FP32-accumulator variant, and to 447.9 MB for the native-BF16 Triton path ($-88\%$ vs reference), which passes the parity test of Table 24. The reduction enables encoder-scale backward at batch sizes that would otherwise require gradient checkpointing. The deltas trace a three-operating-point Pareto frontier between accumulator precision and peak footprint: kernel fusion of the per-level interpolation chain (which the graph-recorded reference

Table 7: Peak GPU memory (MB) at *encoder* scale (A100-SXM4-40GB, BF16, $B=2$ 1536×2048, $L=4$, forward+backward). Reduction is relative to the reference. Best peak in **bold**.

| Backend | Peak (MB) | Reduction |
|---|---|---|
| Reference | 3607.8 | — |
| CUDA *ours* | 1055.9 | −71% |
| Triton, FP32-accumulator | 575.9 | −84% |
| Triton, query-block (native BF16) | **447.9** | −88% |

Table 8: End-to-end MSDA impact at the models' operating points (BF16, pretrained weights). *frac* is MSDA's measured share of the full forward / backward wallclock. The step-speedup columns compare the optimized kernels against four baselines: eager execution (*eag*), whole-model graph compilation (*cmp*), the compiled CUDA kernel shipped by Transformers (Wolf et al., 2020) (*hfc*, the production baseline), and the CUDA kernel of Zhu et al. (2021) (*ref*). The *ref* column is the meaningful backward reference. The *eag*/*cmp* BF16 backward is anomalously slow on A100 (scattered-write contention). Section 5 interprets the architecture crossover and the FP32-accumulator variant.

| Model | GPU | Kernel | MSDA Frac | | Fwd step × | | | | Bwd step × | | | |
|---|---|---|---|---|---|---|---|---|---|---|---|---|
| | | | Fwd | Bwd | eag | cmp | hfc | ref | eag | cmp | hfc | ref |
| Deformable-DETR | A100 | Triton | 0.65 | 0.96 | 1.94 | 1.80 | 1.91 | 1.18 | 2.95 | 2.97 | 2.89 | 0.10 |
| | | CUDA *ours* | | | 1.97 | 1.83 | 1.93 | 1.19 | 31.79 | 31.96 | 31.08 | 1.04 |
| Deformable-DETR | H100 | Triton | 0.63 | 0.78 | 1.92 | 1.74 | 1.91 | 1.18 | 3.41 | 3.40 | 3.46 | 1.33 |
| | | CUDA *ours* | | | 1.93 | 1.74 | 1.91 | 1.18 | 2.59 | 2.58 | 2.63 | 1.01 |
| Mask2Former | A100 | Triton | 0.34 | 0.95 | 1.36 | 1.21 | 1.30 | 1.10 | 2.71 | 2.65 | 2.70 | 0.28 |
| | | CUDA *ours* | | | 1.36 | 1.21 | 1.30 | 1.10 | 9.80 | 9.60 | 9.75 | 1.02 |
| Mask2Former | H100 | Triton | 0.32 | 0.61 | 1.38 | 1.21 | 1.35 | 1.09 | 2.19 | 1.88 | 2.21 | 1.23 |
| | | CUDA *ours* | | | 1.36 | 1.20 | 1.34 | 1.08 | 1.75 | 1.51 | 1.77 | 0.98 |

materialises end-to-end) governs the reference-to-CUDA step, while the accumulator-dtype choice governs the CUDA-to-Triton step (Section 3.3). Per-allocation memory profiling on the A100 partition at the same encoder cell confirms the cross-backend deltas: 1334 MB for the CUDA backend, 663 MB for our query-block Triton backend (a 671 MB kernel-driven saving), and a 129 MB addition for the FP32-accumulator variant (toggling the standard linear-algebra workspace allocator shifts peaks by <3 MB, ruling out caching-allocator artifacts). Table 6 reports the decoder decomposition.

**Roofline analysis (Figure 5).** Every backend on every GPU is memory-bandwidth-bound: the highest AI we measure is 15.1 FLOP/byte (Triton point-parallel backward at FP32 on H100), below the H100 FP32 ridge of ∼17.9 FLOP/byte. FP32 bandwidth utilization is broadly consistent across backends (26–42% of peak on A100, 19–30% on H100), but at BF16 the A100 Triton query-block backward collapses to 2.5% of peak while the CUDA path holds 23.1%; on H100 both backends recover above 19%. This is the direct Roofline signature of the hardware-gated analysis of Section 3.3.

**Technique ablation.** A cumulative technique ablation at encoder scale (Table 25) isolates the kernel-design contributions: on the CUDA path the read-side optimizations (vectorized loads, warp-broadcast coordinates) deliver the majority of the 8× forward improvement, while the write-side relaxed atomics add 7%; the tiling strategy is the dominant Triton lever (8.2× forward).

**End-to-end model impact (Table 8).** To answer whether the operator speedup survives at the model level, the end-to-end study swaps the optimized kernels into stock Transformers (Wolf et al., 2020) Deformable DETR and Mask2Former (pretrained weights) and times the full forward and backward step. The comparison covers eager execution, the same model under whole-model graph compilation (graph capture followed by fusion of

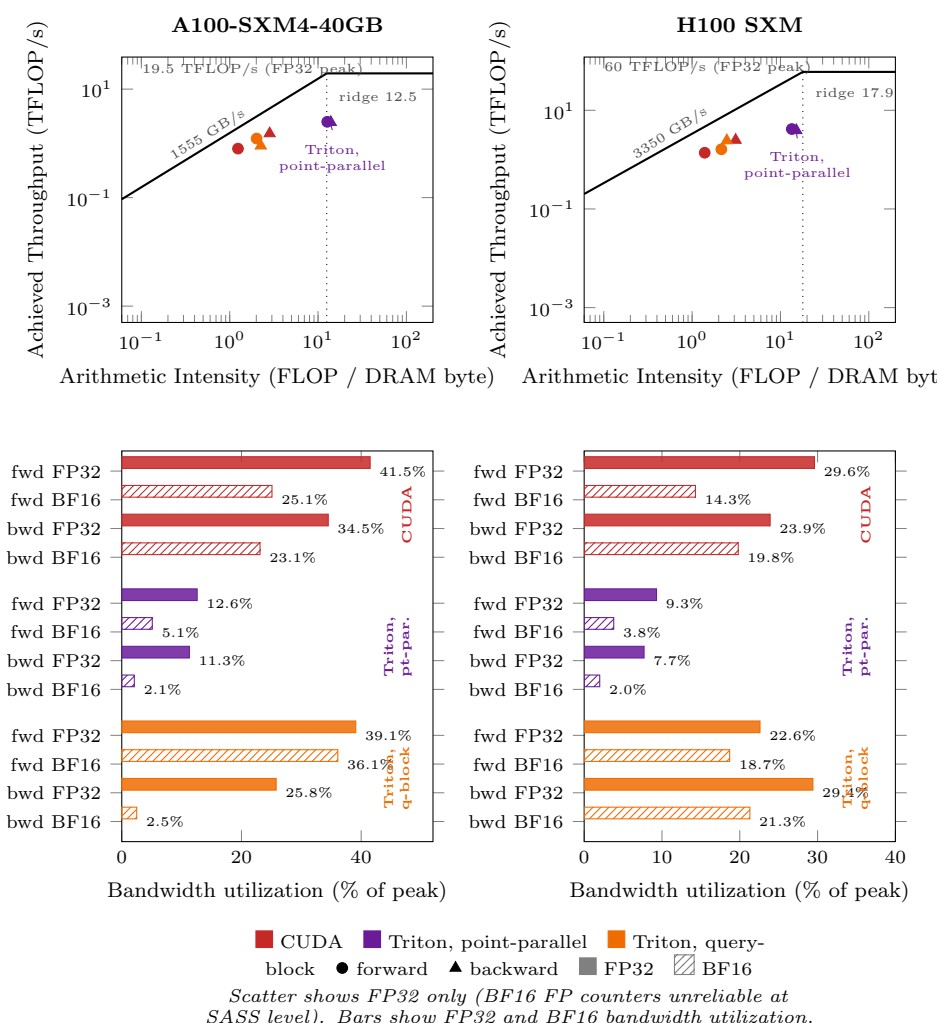

Figure 5: Roofline and bandwidth-utilization summary at decoder scale. *Top:* FP32 arithmetic intensity versus sustained TFLOP/s; all backends fall below the ridge, confirming that MSDA is memory-bandwidth-bound on both GPUs. *Bottom:* bandwidth utilization (% of peak) at FP32 and BF16. The BF16 backward collapse on A100 is the atomic-contention effect of Section 3.3. BF16 arithmetic intensity is omitted because the SASS floating-point counters are unreliable on SM 8.0 at BF16.

the dense layers surrounding MSDA into larger kernels), and the CUDA kernel of Zhu et al. (2021). MSDA is 32 % to 65 % of the forward and 61 % to 96 % of the backward wallclock, so the kernel choice moves the whole step. At the Deformable DETR operating point the Triton kernel cuts the BF16 forward step by 1.9–2.0× over eager execution and 1.7–1.8× over the graph-compiled model. Graph compilation accelerates only the dense surround. At the operator it is *not* faster than eager (Table 15 of Section A) because the scattered-gather operator does not fuse, so the kernel retains its margin over a compiled model. The backward exhibits the same architecture-dependent crossover as the operator study. Triton wins on H100 while CUDA wins on A100 (Section 3.3). An operating-point sweep at the true COCO and Cityscapes resolutions ($Q \approx 22$ to 44k encoder tokens, $L$=4 for Deformable DETR and $L$=3 for Mask2Former) and a forward DRAM-throughput pass (Tables 15 and 16 of Section A) confirm the kernels keep their advantage at production scale. At these scales the reference is DRAM-bandwidth-bound (up to 94 % of peak) while the Triton kernel runs at 91 % to 93 % L2 hit rate across every point, having relocated the bottleneck from HBM into cache (the CUDA kernel relocates partially, 77 % to 93 %). Table 18 of Section A projects per-epoch training wallclock from the measured step times.

Table 9: Captured-replay operating-point comparison (BF16). Every implementation replays the *identical* tensors captured from a pretrained Mask2Former running on real COCO / Cityscapes (CS) images, so the sampling locality is the model's own. Forward / backward are median latency over the captured samples ($p_{10}$–$p_{90}$ spread in brackets); Fwd × is forward speedup over the CUDA kernel of Zhu et al. (2021). Err. is the maximum relative error of the forward output against an FP32 ground truth (0: within BF16-scale tolerance).

| Point ($N$) | Implementation | Fwd (ms) | Bwd (ms) | Fwd (×) | Err. |
|---|---|---|---|---|---|
| M2F/COCO (21504) *A100* | CUDA (Zhu et al., 2021) | 1.91 [1.84–2.01] | 4.09 | 1.0 | 1.33 |
| | Transformers (Wolf et al., 2020) (CUDA) | 1.66 [1.56–1.70] | 22.43 | 1.1 | 1.33 |
| | CUDA *ours* | 0.46 [0.44–0.55] | 3.62 | 4.1 | 0 |
| | Triton, query-block | 0.65 [0.45–0.68] | 50.36 | 2.9 | 0 |
| M2F/CS (43008) *A100* | CUDA (Zhu et al., 2021) | 3.72 [3.60–3.76] | 8.22 | 1.0 | 266 |
| | Transformers (Wolf et al., 2020) (CUDA) | 3.17 [3.09–3.20] | 23.66 | 1.2 | 266 |
| | CUDA *ours* | 0.91 [0.80–0.95] | 7.14 | 4.1 | 0 |
| | Triton, query-block | 0.86 [0.68–0.88] | 95.15 | 4.3 | 0 |
| M2F/COCO (21504) *H100* | CUDA (Zhu et al., 2021) | 1.30 [1.29–1.31] | 2.20 | 1.0 | 1.33 |
| | Transformers (Wolf et al., 2020) (CUDA) | 1.05 [1.04–1.06] | 5.58 | 1.2 | 1.33 |
| | CUDA *ours* | 0.30 [0.30–0.30] | 2.18 | 4.3 | 0 |
| | Triton, query-block | 0.30 [0.30–0.30] | 0.83 | 4.3 | 0 |
| M2F/CS (43008) *H100* | CUDA (Zhu et al., 2021) | 2.51 [2.49–2.53] | 4.40 | 1.0 | 266 |
| | Transformers (Wolf et al., 2020) (CUDA) | 2.04 [2.03–2.05] | 10.56 | 1.2 | 266 |
| | CUDA *ours* | 0.52 [0.51–0.52] | 4.31 | 4.8 | 0 |
| | Triton, query-block | 0.42 [0.42–0.42] | 1.63 | 6.0 | 0 |

**Captured-replay comparison on real model inputs.** The operator timings above feed every kernel synthetic inputs, yet a deformable gather is dominated by its sampling *locality*. Uniform-random sampling locations (Table 17) thrash DRAM, whereas a trained model samples in tight, cache-friendly clusters around its reference points. To compare implementations on the distribution they actually run at, the captured-replay study records the exact tensors (the input feature map, sampling locations, and attention weights) a pretrained Mask2Former feeds each of its six pixel-decoder MSDA layers over real COCO and Cityscapes images, and replays the *identical* captured tensors through every implementation (Table 9). Even at this realistic locality, which is comparatively favorable for the simpler Zhu et al. (2021) kernel, the optimized kernels keep a wide margin. At BF16 the forward runs 2.9 to 6.0× faster than the CUDA kernel of Zhu et al. (2021) and 2.6 to 4.9× faster than the CUDA kernel shipped in Transformers (Wolf et al., 2020), across both GPUs and operating points. On these same inputs the FP32-accumulating kernels match the FP32 reference to tolerance while the BF16-accumulating reference kernels do not (maximum relative error up to 266, error column of Table 9). The backward shows the same architecture-dependent crossover as the operator study (Section 3.3). Triton is fastest on H100 but its A100 atomics regress, where CUDA is the production backward. The $p_{10}$–$p_{90}$ spread (mostly within a few percent across 150 captured samples per point) shows the margin is not an artifact of any single image.

## 6 Related Work

**Deformable operators.** We take the MSDA algorithm as given and ask a systems question about it. The algorithm traces back to DCN's learnable 2D offsets (Dai et al., 2017; Zhu et al., 2019; Wang et al., 2023), which Zhu et al. (2021) transposed to attention, replacing DETR's dense $O(N^2)$ matmul (Carion et al., 2020) with a

small fixed number of bilinearly sampled points per query; MSDA is now a core component of DINO (Zhang et al., 2023) and Mask2Former (Cheng et al., 2022). The closest systems-level work is DCNv4 (Xiong et al., 2024), which accelerates deformable *convolution*. It reports $3\times$ over DCNv3 through vectorized 128-bit loads, thread-to-channel remapping within a fixed $3\times3$ deformable-convolution footprint, and half-precision arithmetic (DCNv4 Table 10 indicates that the remapping in isolation is mildly anti-helpful; the speedup materializes only with vectorization and FP16 together). DCNv4 §3.2 states that these optimizations also apply to deformable attention, but the published benchmarks are on deformable convolution rather than MSDA. DCNv4's other headline change, removing the softmax normalization that DCNv3 (Wang et al., 2023) introduced over sampling points, alters the operator rather than its implementation. Carried to attention, the aggregation weights would no longer satisfy the sum-to-one constraint that defines MSDA (Section 2.1), so a multi-scale variant would be a different operator, not a faster MSDA. Both levers that ablation credits are already standard practice in the kernels studied here rather than a comparison baseline. Vectorized 128-bit loads enter with the first-pass optimizations of Section 2.4 (Table 25 isolates their contribution), and half-precision arithmetic is central to the entire study. The remapping is bound to the fixed convolution footprint, which has no analogue in the data-dependent multi-scale sampling of MSDA. The measured MSDA baselines are the published kernels of Table 17 and Zhu et al. (2021).

**Dense attention kernels.** FlashAttention (Dao, 2024; Shah et al., 2024) and FlashInfer (Ye et al., 2025) achieve substantial speedups on dense self-attention by tiling $QK^\top V$ to keep the attention matrix in SRAM; FlashAttention-3 adds Hopper warp specialization. FlexAttention (Dong et al., 2024) generalizes tiled attention to user-defined score modifications. Their tile structure does not transfer to MSDA, whose per-query learned offsets vary the $LK$ sampled positions and so admit no static tiling. The production attention libraries built on these kernels (for example xFormers (Lefaudeux et al., 2022)) correspondingly expose dense and block-sparse self-attention, not a deformable-attention operator, so the de-facto MSDA reference remains the original CUDA kernel of Zhu et al. (2021) and its native-PyTorch bilinear-sampling fallback. Table 17 benchmarks these shipped implementations directly. At encoder scale the Triton kernel is $3$–$5\times$ faster than Transformers' (Wolf et al., 2020) compiled CUDA MSDA (up to $5.1\times$ at the Cityscapes encoder operating point), the production baseline. Against its native-PyTorch bilinear-sampling fallback, which is rarely active when CUDA is available, the margin widens to $11$–$21\times$. The difference is access pattern, not algorithm. Dense matmul attention is compute-bound at large $N$, whereas MSDA is memory-bandwidth-bound throughout (Figure 5), its cost dominated by scattered irregular reads. A same-task substitution of dense attention for MSDA is, moreover, not a drop-in baseline but a different model (the DETR-to-Deformable-DETR step (Carion et al., 2020; Zhu et al., 2021)): a per-head $O(N^2)$ score matrix over the encoder's $N$ ($\approx 22$k for COCO, $\approx 44$k for Cityscapes) flattened feature tokens is compute-bound regardless of tiling. Even FlashAttention, which keeps that matrix in SRAM, remains $O(N^2)$ and runs $12$–$15\times$ slower than MSDA at COCO scale and $29$–$38\times$ slower at Cityscapes encoder scale (Table 17), which is why MSDA replaces full attention at the encoder.

**MSDA-specific acceleration.** DEFA (Xu et al., 2024) combines algorithm-level pruning with a custom ASIC/FPGA for $10$–$32\times$ speedup; its GPU-side profiling independently confirms the memory-boundedness diagnosis (multi-scale sampling and aggregation account for over $60\,\%$ of MSDA inference time on an RTX 3090 Ti despite only $3.25\,\%$ of FLOPs). Huang et al. (2025) achieve $5.9\times$ forward / $8.9\times$ backward on Huawei Ascend NPUs via staggered gradient computation, complementary to our relaxed-ordering atomics. DANMP (Li et al., 2026) places compute units adjacent to HBM banks for $97\times$ over an A6000 baseline. Their GPU-side profiling (DANMP §3.1, RTX A6000) reports the MSDA cache hit rate declining monotonically as batch size grows and attributes the effect to data-dependent targets across batches preventing conventional locality-based reuse. DANMP also proposes a software-side *Clustering-and-Packing* (CAP) reordering scheme (§6.3) that achieves a $1.45\times$ speedup on CPU; CAP is not evaluated on GPU in that work, a gap we address with a direct GPU analogue in Section 3.1. UEDA (Sun et al., 2025) and QUILL (Oh et al., 2026) are ASICs for deformable-attention variants; QUILL's dynamic query-order optimization overlaps reordering with compute in hardware where our null result on commodity GPUs would otherwise apply. We target NVIDIA CUDA and Triton, provide a cross-paradigm comparison across two GPU generations, and document a locality negative result specific to the GPU execution model.

**Locality reordering in sparse operators.** Our Morton Z-order null (Section 3.1) echoes mixed GNN experience with vertex reordering (Merkel et al., 2025; Wang et al., 2021) and the structured-pattern conclusion of Yuan et al. (2025) for LLM sparse attention; space-filling-curve reordering *does* succeed when the tokens inhabit a fixed geometric substrate (HilbertA (Zheng et al., 2025) on diffusion transformers, Point Transformer V3 (Wu et al., 2024) on point clouds), a precondition MSDA's learned, per-iteration sampling does not meet.

**Atomic optimization.** Concurrent work on differentiable rendering reduces atomic contention through warp-level reduction primitives (Durvasula et al., 2024; 2025; Mallick et al., 2024) and queueing-theoretic modelling (Dong & Pai, 2025); Section 3.3 complements these by quantifying the magnitude on MSDA and locating the architectural close at the SM 8.0–SM 9.0 boundary.

**The Triton ecosystem.** Triton (Tillet et al., 2019) is widely adopted for LLM inference and dense attention (Dao et al., 2022; Lefaudeux et al., 2022; Hsu et al., 2025), all of which exploit contiguous $QK^\top V$ tile structure that MSDA lacks. The scattered-access comparison here finds an architecture-dependent verdict (Triton wins the forward pass via query-dimension tiling, while the backward winner is gated by the target atomic hardware) rather than a paradigm-dependent one.

## 7  Discussion and Conclusion

We characterized how scattered-access attention operators interact with the conventional GPU optimization toolkit and found two failure modes and one hardware-gated root cause. Dispatch-order reordering does not pay for MSDA. Across sampling-point counts $K \in \{4, 8, 16\}$ and level counts $L \in \{2, 3, 4, 5\}$ at the decoder operating point, the seven query-dispatch orders, including a faithful GPU port of DANMP's K-means clustering substep, are statistically indistinguishable from linear, because L2 locality is tile-set by the query-block kernel, not by the global dispatch order (Section 3.1). High occupancy is not high throughput either: an 85 %-occupancy tiling delivers 5.1 % of A100 peak bandwidth where a 17 %-occupancy tiling achieves 36 % and runs 7.4× faster (Section 3.2). The backward BF16 kernel then collapses to 2.5 % of A100 peak bandwidth because SM 8.0 lacks a native BF16 atomic-add primitive and the CUDA runtime emits a CAS loop, while the same code recovers to 21.3 % on H100 (Section 3.3). An FP32-accumulator variant closes the A100 gap, making the backward-pass choice an *accumulator-precision* decision gated by target hardware (Section 3.3 and Table 4).

**Lessons for practitioners.** Four scoped hypotheses follow, to be tested on other scattered-access operators: (i) do not sort the query dispatch order by space-filling curves without a tile-set exploitability check; (ii) do not equate high occupancy with high throughput on memory-bound operators, tuning instead for register reuse and wide DRAM transactions; (iii) avoid native half-precision atomic addition on architectures predating dedicated reduction hardware, substituting FP32 accumulators or compiler-controlled relaxed atomics; (iv) profile DRAM bandwidth, not FLOPs, as the roofline axis for memory-bound operators. All three failure modes motivating (i)–(iii) were invisible to FLOP-based metrics.

**Limitations and future work.** The backward atomic bottleneck is a hardware-*class* property, gated by whether an architecture exposes a native low-precision atomic, not a per-chip quirk. The two NVIDIA generations tested instantiate both sides of this divide (SM 8.0 without, SM 9.0 with), and the operator-agnostic scatter-add control confirms the pattern outside MSDA. AMD and Intel hardware would test the same boundary. TensorRT's deformable-attention plugin is inference-only, precluding a training-time comparison, and we do not report end-to-end training curves. We have, however, verified inference-time task parity by swapping our backends into released Deformable DETR and Mask2Former checkpoints and reproducing their reported COCO, Cityscapes, and ADE20K metrics within rounding (Table 9, no end-to-end retraining). The native-BF16 backward's relaxed-ordering atomics do make training run-to-run nondeterministic. Table 10 bounds the gradient change at $4 \times 10^{-5}$–$3 \times 10^{-4}$ relative $L_2$ across runs, four orders of magnitude below the $\approx 1.4$ variation between minibatches, so it does not affect convergence. The FP32-accumulator variant removes it entirely, trading a 29 % larger backward buffer for faster atomics. A future driver or SRAM-based

Table 10: Gradient nondeterminism of the relaxed-ordering BF16 backward, on synthetic operating-point inputs ($n$=5 repeats). $\epsilon_{\text{atomic}}$ is the run-to-run relative-$L_2$ change of the input gradient on the *same* input; $\epsilon_{\text{minibatch}}$ is the relative-$L_2$ change between two *different* inputs (the variation SGD already sees).

| | $\epsilon_{\text{atomic}}$ | | | |
|---|---|---|---|---|
| Operating point | A100 | H100 | $\epsilon_{\text{minibatch}}$ | Ratio |
| DETR / COCO | $2.0 \times 10^{-4}$ | $2.6 \times 10^{-4}$ | 1.4 | $1.4 \times 10^{-4}$ |
| DETR / Cityscapes | $1.1 \times 10^{-4}$ | $1.4 \times 10^{-4}$ | 1.4 | $7.8 \times 10^{-5}$ |
| M2F / COCO | $6.4 \times 10^{-5}$ | $7.8 \times 10^{-5}$ | 1.4 | $4.7 \times 10^{-5}$ |
| M2F / Cityscapes | $3.9 \times 10^{-5}$ | $4.9 \times 10^{-5}$ | 1.4 | $2.8 \times 10^{-5}$ |

Triton backward could narrow the gap without the memory cost. We will release our implementation upon acceptance.

**Broader impact.** This work improves the efficiency of an existing operator. It adds no new model capability and opens no new avenue for misuse. Its main externality is reduced compute and energy for training and serving DETR-family models. Per-epoch training wallclock drops by 5–20% depending on model and GPU (Table 18), and the 88 % lower peak memory (Section 5) fits the same workloads onto smaller accelerators. Because the kernels match the reference to floating-point tolerance on the inputs a trained model actually produces (Tables 9 and 24), deployed accuracy and safety behavior are unchanged.

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

## A    Full Benchmark Tables

Table 11 reports the full per-cell dispatch-order measurements summarized in Section 3.1: $p_{50}$ latency, per-cell standard deviation $\sigma$, relative standard deviation $\sigma/\mu$, and ratio $r$ vs. the (config, dtype) linear baseline, for all eight dispatch orders across the three decoder batch sizes and three precisions on both GPUs ($n = 100$ trials per cell). Cells exceeding nvbench's 0.5% relative- standard-deviation reproducibility floor are flagged with †. The shared compute nodes used in our measurements are not externally clock-locked, so per-trial $\sigma/\mu$ is typically in the 1–3% band rather than the 0.5% nvbench-locked floor; the $n = 100$-trial mean is nevertheless discriminating to a much tighter level (see methods footnote in Section 3.1).

Table 11: Per-cell locality-controls measurements ($n = 100$ trials each). $r$ is the ratio of the cell's $p_{50}$ latency to its (config, dtype, batch) linear baseline, with a 95% confidence interval propagated from the reported $\sigma$ (normal approximation, $\text{SE} = \sigma/\sqrt{n}$). †: $\sigma/\mu$ exceeds nvbench's 0.5% reproducibility floor. ‡: $p_{99} \geq 1.5\,p_{50}$ (run-to-run tail-spike contamination). $r$ is undetermined for these cells and they are excluded from the ±2% null.

| GPU | Config | B | Order | $p_{50}$ (ms) | $p_{99}$ (ms) | $\sigma/\mu$ | $r$ | CI$_{95}$ |
|-----|--------|---|-------|------|------|------|------|------|
| A100 | B1/FP32 | 1 | linear | 0.5212 | 0.6164 | 2.57%† | 1.0000 | — |
| A100 | B1/FP32 | 1 | Morton | 0.5233 | 0.5960 | 2.45%† | 1.0040 | ±0.0070 |
| A100 | B1/FP32 | 1 | random | 0.5202 | 0.5448 | 1.35%† | 0.9981 | ±0.0057 |
| A100 | B1/FP32 | 1 | scanline | 0.5192 | 0.5468 | 1.44%† | 0.9962 | ±0.0058 |
| A100 | B1/FP32 | 1 | centroid | 0.5217 | 0.5663 | 1.84%† | 1.0010 | ±0.0062 |
| A100 | B1/FP32 | 1 | Hilbert | 0.5192 | 0.5458 | 1.44%† | 0.9962 | ±0.0058 |
| A100 | B1/FP32 | 1 | K-means CAP | 0.5181 | 0.5847 | 1.97%† | 0.9941 | ±0.0063 |
| A100 | B1/FP32 | 1 | partial DANMP | 0.5192 | 0.5448 | 1.54%† | 0.9962 | ±0.0059 |
| A100 | B1/FP16 | 1 | linear | 0.5550 | 0.5929 | 1.42%† | 1.0000 | — |
| A100 | B1/FP16 | 1 | Morton | 0.5550 | 0.5929 | 1.41%† | 1.0000 | ±0.0039 |
| A100 | B1/FP16 | 1 | random | 0.5540 | 0.5929 | 1.37%† | 0.9982 | ±0.0039 |
| A100 | B1/FP16 | 1 | scanline | 0.5540 | 0.5724 | 1.17%† | 0.9982 | ±0.0036 |
| A100 | B1/FP16 | 1 | centroid | 0.5540 | 0.5704 | 1.06%† | 0.9982 | ±0.0035 |
| A100 | B1/FP16 | 1 | Hilbert | 0.5591 | 0.5888 | 2.06%† | 1.0074 | ±0.0049 |
| A100 | B1/FP16 | 1 | K-means CAP | 0.5652 | 0.6185 | 2.12%† | 1.0184 | ±0.0051 |
| A100 | B1/FP16 | 1 | partial DANMP | 0.5642 | 0.6134 | 2.41%† | 1.0166 | ±0.0056 |
| A100 | B1/BF16 | 1 | linear | 0.5386 | 0.5796 | 1.71%† | 1.0000 | — |
| A100 | B1/BF16 | 1 | Morton | 0.5376 | 0.5857 | 2.19%† | 0.9981 | ±0.0054 |
| A100 | B1/BF16 | 1 | random | 0.5386 | 0.5775 | 1.75%† | 1.0000 | ±0.0048 |
| A100 | B1/BF16 | 1 | scanline | 0.5345 | 0.5724 | 1.81%† | 0.9924 | ±0.0048 |
| A100 | B1/BF16 | 1 | centroid | 0.5386 | 0.5980 | 2.04%† | 1.0000 | ±0.0052 |
| A100 | B1/BF16 | 1 | Hilbert | 0.5417 | 0.5990 | 2.34%† | 1.0058 | ±0.0057 |
| A100 | B1/BF16 | 1 | K-means CAP | 0.5366 | 0.5765 | 1.88%† | 0.9963 | ±0.0050 |
| A100 | B1/BF16 | 1 | partial DANMP | 0.5376 | 0.5714 | 1.99%† | 0.9981 | ±0.0051 |
| A100 | B2/enc/BF16 | 2 | linear | 2.0029 | 2.0644 | 0.31% | 1.0000 | — |
| A100 | B2/enc/BF16 | 2 | Morton | 1.9948 | 2.0521 | 0.30% | 0.9960 | ±0.0008 |

*(continued on next page)*

*(Table 11 continued from previous page.)*

| GPU | Config | B | Order | $p_{50}$ (ms) | $p_{99}$ (ms) | $\sigma/\mu$ | $r$ | CI$_{95}$ |
|---|---|---|---|---|---|---|---|---|
| A100 | B2/enc/BF16 | 2 | random | 2.0029 | 2.0572 | 0.28% | 1.0000 | ±0.0008 |
| A100 | B2/enc/BF16 | 2 | scanline | 1.9937 | 2.0572 | 0.38% | 0.9954 | ±0.0010 |
| A100 | B2/enc/BF16 | 2 | centroid | 2.0040 | 2.0603 | 0.29% | 1.0005 | ±0.0008 |
| A100 | B2/enc/BF16 | 2 | Hilbert | 1.9948 | 2.0511 | 0.29% | 0.9960 | ±0.0008 |
| A100 | B2/enc/BF16 | 2 | K-means CAP | 2.0029 | 2.0593 | 0.29% | 1.0000 | ±0.0008 |
| A100 | B2/enc/BF16 | 2 | partial DANMP | 2.0029 | 2.0593 | 0.29% | 1.0000 | ±0.0008 |
| A100 | B2/FP32 | 2 | linear | 0.5202 | 0.5652 | 2.06%† | 1.0000 | — |
| A100 | B2/FP32 | 2 | Morton | 0.5197 | 0.5652 | 1.77%† | 0.9990 | ±0.0053 |
| A100 | B2/FP32 | 2 | random | 0.5202 | 0.6963 | 3.73%† | 1.0000 | ±0.0083 |
| A100 | B2/FP32 | 2 | scanline | 0.5181 | 0.5427 | 1.49%† | 0.9960 | ±0.0050 |
| A100 | B2/FP32 | 2 | centroid | 0.5176 | 0.6267 | 2.51%† | 0.9950 | ±0.0063 |
| A100 | B2/FP32 | 2 | Hilbert | 0.5171 | 0.5632 | 1.84%† | 0.9940 | ±0.0054 |
| A100 | B2/FP32 | 2 | K-means CAP | 0.5222 | 0.5755 | 1.86%† | 1.0038 | ±0.0055 |
| A100 | B2/FP32 | 2 | partial DANMP | 0.5212 | 0.5765 | 2.51%† | 1.0019 | ±0.0064 |
| A100 | B2/FP16 | 2 | linear | 0.5509 | 0.6052 | 2.40%† | 1.0000 | — |
| A100 | B2/FP16 | 2 | Morton | 0.5427 | 0.6216 | 2.19%† | 0.9851 | ±0.0063 |
| A100 | B2/FP16 | 2 | random | 0.5417 | 0.5724 | 1.14%† | 0.9833 | ±0.0051 |
| A100 | B2/FP16 | 2 | scanline | 0.5407 | 0.5581 | 1.07%† | 0.9815 | ±0.0051 |
| A100 | B2/FP16 | 2 | centroid | 0.5422 | 0.5652 | 1.14%† | 0.9842 | ±0.0051 |
| A100 | B2/FP16 | 2 | Hilbert | 0.5417 | 0.6072 | 2.64%† | 0.9833 | ±0.0069 |
| A100 | B2/FP16 | 2 | K-means CAP | 0.5417 | 0.5878 | 1.68%† | 0.9833 | ±0.0056 |
| A100 | B2/FP16 | 2 | partial DANMP | 0.5417 | 0.5601 | 1.13%† | 0.9833 | ±0.0051 |
| A100 | B2/BF16 | 2 | linear | 0.5212 | 0.5683 | 1.65%† | 1.0000 | — |
| A100 | B2/BF16 | 2 | linear | 0.5335 | 0.6072 | 2.25%† | 1.0000 | — |
| A100 | B2/BF16 | 2 | Morton | 0.5222 | 0.5478 | 1.40%† | 0.9788 | ±0.0051 |
| A100 | B2/BF16 | 2 | Morton | 0.5366 | 0.6124 | 2.24%† | 1.0058 | ±0.0063 |
| A100 | B2/BF16 | 2 | random | 0.5217 | 0.5652 | 1.44%† | 0.9779 | ±0.0051 |
| A100 | B2/BF16 | 2 | random | 0.5356 | 0.5806 | 1.59%† | 1.0039 | ±0.0054 |
| A100 | B2/BF16 | 2 | scanline | 0.5228 | 0.5612 | 1.34%† | 0.9799 | ±0.0050 |
| A100 | B2/BF16 | 2 | scanline | 0.5325 | 0.5550 | 1.24%† | 0.9981 | ±0.0050 |
| A100 | B2/BF16 | 2 | centroid | 0.5228 | 0.5366 | 1.03%† | 0.9799 | ±0.0048 |
| A100 | B2/BF16 | 2 | centroid | 0.5356 | 0.5652 | 1.44%† | 1.0039 | ±0.0053 |
| A100 | B2/BF16 | 2 | Hilbert | 0.5222 | 0.5540 | 1.42%† | 0.9788 | ±0.0051 |
| A100 | B2/BF16 | 2 | Hilbert | 0.5345 | 0.5601 | 1.44%† | 1.0019 | ±0.0052 |
| A100 | B2/BF16 | 2 | K-means CAP | 0.5217 | 0.5356 | 1.07%† | 0.9779 | ±0.0048 |
| A100 | B2/BF16 | 2 | K-means CAP | 0.5376 | 0.5775 | 1.56%† | 1.0077 | ±0.0054 |
| A100 | B2/BF16 | 2 | partial DANMP | 0.5222 | 0.5458 | 1.24%† | 0.9788 | ±0.0049 |
| A100 | B2/BF16 | 2 | partial DANMP | 0.5366 | 0.6093 | 2.05%† | 1.0058 | ±0.0060 |
| A100 | B4/FP32 | 4 | linear | 0.9723 | 1.0230 | 1.39%† | 1.0000 | — |
| A100 | B4/FP32 | 4 | Morton | 0.9754 | 1.0281 | 1.21%† | 1.0032 | ±0.0036 |
| A100 | B4/FP32 | 4 | random | 0.9769 | 1.0250 | 1.36%† | 1.0047 | ±0.0038 |
| A100 | B4/FP32 | 4 | scanline | 0.9748 | 1.0322 | 1.30%† | 1.0026 | ±0.0037 |
| A100 | B4/FP32 | 4 | centroid | 0.9748 | 1.0363 | 1.31%† | 1.0026 | ±0.0038 |
| A100 | B4/FP32 | 4 | Hilbert | 0.9779 | 1.0373 | 1.38%† | 1.0058 | ±0.0039 |
| A100 | B4/FP32 | 4 | K-means CAP | 0.9733 | 1.0322 | 1.41%† | 1.0010 | ±0.0039 |
| A100 | B4/FP32 | 4 | partial DANMP | 0.9728 | 1.0127 | 1.22%† | 1.0005 | ±0.0036 |
| A100 | B4/FP16 | 4 | linear | 0.5417 | 1.0271 | 9.40%† | 1.0000 | — |
| A100 | B4/FP16 | 4 | Morton | 0.5478 | 0.5827 | 2.34%† | 1.0113 | ±0.0192 |
| A100 | B4/FP16 | 4 | random | 0.5432 | 0.6083 | 2.65%† | 1.0028 | ±0.0192 |
| A100 | B4/FP16 | 4 | scanline | 0.5376 | 0.5888 | 2.62%† | 0.9924 | ±0.0190 |
| A100 | B4/FP16 | 4 | centroid | 0.5519 | 0.5949 | 2.90%† | 1.0188 | ±0.0196 |
| A100 | B4/FP16 | 4 | Hilbert | 0.5535 | 0.5980 | 2.80%† | 1.0218 | ±0.0196 |
| A100 | B4/FP16 | 4 | K-means CAP | 0.5468 | 0.6185 | 2.83%† | 1.0094 | ±0.0194 |
| A100 | B4/FP16 | 4 | partial DANMP | 0.5458 | 0.5878 | 3.00%† | 1.0076 | ±0.0195 |
| A100 | B4/BF16 | 4 | linear | 0.5304 | 0.5960 | 3.02%† | 1.0000 | — |
| A100 | B4/BF16 | 4 | Morton | 0.5335 | 0.5642 | 2.55%† | 1.0058 | ±0.0078 |
| A100 | B4/BF16 | 4 | random | 0.5192 | 0.5775 | 2.48%† | 0.9789 | ±0.0075 |
| A100 | B4/BF16 | 4 | scanline | 0.5171 | 0.5396 | 1.22%† | 0.9749 | ±0.0062 |
| A100 | B4/BF16 | 4 | centroid | 0.5161 | 0.5376 | 1.16%† | 0.9730 | ±0.0062 |
| A100 | B4/BF16 | 4 | Hilbert | 0.5181 | 0.6113 | 2.57%† | 0.9768 | ±0.0076 |
| A100 | B4/BF16 | 4 | K-means CAP | 0.5161 | 0.5448 | 1.14%† | 0.9730 | ±0.0062 |
| A100 | B4/BF16 | 4 | partial DANMP | 0.5161 | 2.8744 | 45.73%† | 0.9730‡ | n/d |
| A100 | B8/FP32 | 8 | linear | 1.4991 | 1.5555 | 0.61%† | 1.0000 | — |
| A100 | B8/FP32 | 8 | Morton | 1.4991 | 1.5340 | 0.49% | 1.0000 | ±0.0015 |

*(continued on next page)*

*(Table 11 continued from previous page.)*

| GPU | Config | B | Order | $p_{50}$ (ms) | $p_{99}$ (ms) | $\sigma/\mu$ | $r$ | CI$_{95}$ |
|---|---|---|---|---|---|---|---|---|
| A100 | B8/FP32 | 8 | random | 1.4991 | 1.5473 | 0.58%† | 1.0000 | ±0.0016 |
| A100 | B8/FP32 | 8 | scanline | 1.4981 | 1.5493 | 0.58%† | 0.9993 | ±0.0016 |
| A100 | B8/FP32 | 8 | centroid | 1.4981 | 1.5514 | 0.63%† | 0.9993 | ±0.0017 |
| A100 | B8/FP32 | 8 | Hilbert | 1.4991 | 1.5514 | 0.62%† | 1.0000 | ±0.0017 |
| A100 | B8/FP32 | 8 | K-means CAP | 1.4981 | 1.5421 | 0.57%† | 0.9993 | ±0.0016 |
| A100 | B8/FP32 | 8 | partial DANMP | 1.4991 | 1.5544 | 0.60%† | 1.0000 | ±0.0017 |
| A100 | B8/FP16 | 8 | linear | 0.5494 | 0.5745 | 1.18%† | 1.0000 | — |
| A100 | B8/FP16 | 8 | Morton | 0.5489 | 0.5755 | 1.13%† | 0.9991 | ±0.0032 |
| A100 | B8/FP16 | 8 | random | 0.5489 | 0.5724 | 1.09%† | 0.9991 | ±0.0032 |
| A100 | B8/FP16 | 8 | scanline | 0.5489 | 0.5734 | 1.24%† | 0.9991 | ±0.0034 |
| A100 | B8/FP16 | 8 | centroid | 0.5489 | 0.5714 | 1.11%† | 0.9991 | ±0.0032 |
| A100 | B8/FP16 | 8 | Hilbert | 0.5489 | 0.5622 | 0.98%† | 0.9991 | ±0.0030 |
| A100 | B8/FP16 | 8 | K-means CAP | 0.5489 | 0.5868 | 1.28%† | 0.9991 | ±0.0034 |
| A100 | B8/FP16 | 8 | partial DANMP | 0.5668 | 0.6134 | 2.77%† | 1.0317 | ±0.0061 |
| A100 | B8/BF16 | 8 | linear | 0.5248 | 0.5827 | 2.59%† | 1.0000 | — |
| A100 | B8/BF16 | 8 | Morton | 0.5171 | 0.5755 | 2.07%† | 0.9853 | ±0.0064 |
| A100 | B8/BF16 | 8 | random | 0.5151 | 0.5396 | 1.26%† | 0.9815 | ±0.0055 |
| A100 | B8/BF16 | 8 | scanline | 0.5161 | 0.5386 | 1.18%† | 0.9834 | ±0.0055 |
| A100 | B8/BF16 | 8 | centroid | 0.5161 | 0.5427 | 1.16%† | 0.9834 | ±0.0055 |
| A100 | B8/BF16 | 8 | Hilbert | 0.5161 | 0.5530 | 1.43%† | 0.9834 | ±0.0057 |
| A100 | B8/BF16 | 8 | K-means CAP | 0.5161 | 0.5489 | 1.38%† | 0.9834 | ±0.0057 |
| A100 | B8/BF16 | 8 | partial DANMP | 0.5151 | 1.1121 | 11.78%† | 0.9815‡ | n/d |
| H100 | B1/FP32 | 1 | linear | 0.4348 | 0.4998 | 2.46%† | 1.0000 | — |
| H100 | B1/FP32 | 1 | Morton | 0.4348 | 0.4795 | 1.91%† | 1.0000 | ±0.0061 |
| H100 | B1/FP32 | 1 | random | 0.4388 | 0.4571 | 1.80%† | 1.0092 | ±0.0060 |
| H100 | B1/FP32 | 1 | scanline | 0.4328 | 0.4631 | 1.52%† | 0.9954 | ±0.0056 |
| H100 | B1/FP32 | 1 | centroid | 0.4367 | 0.4596 | 1.65%† | 1.0044 | ±0.0058 |
| H100 | B1/FP32 | 1 | Hilbert | 0.4377 | 0.4663 | 1.69%† | 1.0067 | ±0.0059 |
| H100 | B1/FP32 | 1 | K-means CAP | 0.4367 | 0.4595 | 1.72%† | 1.0044 | ±0.0059 |
| H100 | B1/FP32 | 1 | partial DANMP | 0.4367 | 0.4683 | 1.83%† | 1.0044 | ±0.0060 |
| H100 | B1/FP16 | 1 | linear | 0.4157 | 0.4577 | 2.04%† | 1.0000 | — |
| H100 | B1/FP16 | 1 | Morton | 0.4163 | 0.4329 | 1.51%† | 1.0014 | ±0.0050 |
| H100 | B1/FP16 | 1 | random | 0.4156 | 0.4346 | 1.40%† | 0.9998 | ±0.0049 |
| H100 | B1/FP16 | 1 | scanline | 0.4156 | 2.1844 | 42.56%† | 0.9998‡ | n/d |
| H100 | B1/FP16 | 1 | centroid | 0.4142 | 0.4381 | 1.55%† | 0.9964 | ±0.0050 |
| H100 | B1/FP16 | 1 | Hilbert | 0.4141 | 0.4476 | 1.62%† | 0.9962 | ±0.0051 |
| H100 | B1/FP16 | 1 | K-means CAP | 0.4168 | 0.4368 | 1.56%† | 1.0026 | ±0.0051 |
| H100 | B1/FP16 | 1 | partial DANMP | 0.4159 | 0.4396 | 1.76%† | 1.0005 | ±0.0053 |
| H100 | B1/BF16 | 1 | linear | 0.4152 | 0.4509 | 1.73%† | 1.0000 | — |
| H100 | B1/BF16 | 1 | Morton | 0.4176 | 0.4373 | 1.53%† | 1.0058 | ±0.0046 |
| H100 | B1/BF16 | 1 | random | 0.4148 | 0.4348 | 1.52%† | 0.9990 | ±0.0045 |
| H100 | B1/BF16 | 1 | scanline | 0.4170 | 0.4460 | 1.58%† | 1.0043 | ±0.0046 |
| H100 | B1/BF16 | 1 | centroid | 0.4155 | 0.4492 | 1.64%† | 1.0007 | ±0.0047 |
| H100 | B1/BF16 | 1 | Hilbert | 0.4159 | 0.5391 | 3.37%† | 1.0017 | ±0.0074 |
| H100 | B1/BF16 | 1 | K-means CAP | 0.4147 | 0.4471 | 1.66%† | 0.9988 | ±0.0047 |
| H100 | B1/BF16 | 1 | partial DANMP | 0.4172 | 0.4447 | 1.58%† | 1.0048 | ±0.0046 |
| H100 | B2/enc/BF16 | 2 | linear | 1.1508 | 1.2196 | 0.64%† | 1.0000 | — |
| H100 | B2/enc/BF16 | 2 | Morton | 1.1471 | 1.2108 | 0.60%† | 0.9968 | ±0.0017 |
| H100 | B2/enc/BF16 | 2 | random | 1.1532 | 1.2776 | 1.12%† | 1.0021 | ±0.0025 |
| H100 | B2/enc/BF16 | 2 | scanline | 1.1525 | 1.2161 | 0.62%† | 1.0015 | ±0.0017 |
| H100 | B2/enc/BF16 | 2 | centroid | 1.1566 | 1.2158 | 0.55%† | 1.0050 | ±0.0017 |
| H100 | B2/enc/BF16 | 2 | Hilbert | 1.1536 | 1.2181 | 0.59%† | 1.0024 | ±0.0017 |
| H100 | B2/enc/BF16 | 2 | K-means CAP | 1.1561 | 1.2211 | 0.60%† | 1.0046 | ±0.0017 |
| H100 | B2/enc/BF16 | 2 | partial DANMP | 1.1555 | 1.2202 | 0.61%† | 1.0041 | ±0.0018 |
| H100 | B2/FP32 | 2 | linear | 0.4264 | 0.4491 | 1.50%† | 1.0000 | — |
| H100 | B2/FP32 | 2 | Morton | 0.4282 | 0.4533 | 1.73%† | 1.0042 | ±0.0045 |
| H100 | B2/FP32 | 2 | random | 0.4249 | 0.4640 | 1.86%† | 0.9965 | ±0.0047 |
| H100 | B2/FP32 | 2 | scanline | 0.4243 | 0.4440 | 1.58%† | 0.9951 | ±0.0042 |
| H100 | B2/FP32 | 2 | centroid | 0.4244 | 0.4569 | 1.89%† | 0.9953 | ±0.0047 |
| H100 | B2/FP32 | 2 | Hilbert | 0.4253 | 0.4519 | 1.58%† | 0.9974 | ±0.0043 |
| H100 | B2/FP32 | 2 | K-means CAP | 0.4264 | 0.4690 | 1.83%† | 1.0000 | ±0.0046 |
| H100 | B2/FP32 | 2 | partial DANMP | 0.4256 | 0.4566 | 1.72%† | 0.9981 | ±0.0045 |
| H100 | B2/FP16 | 2 | linear | 0.4147 | 0.4428 | 1.62%† | 1.0000 | — |
| H100 | B2/FP16 | 2 | Morton | 0.4148 | 0.4378 | 1.47%† | 1.0002 | ±0.0043 |

*(continued on next page)*

*(Table 11 continued from previous page.)*

| GPU | Config | B | Order | $p_{50}$ (ms) | $p_{99}$ (ms) | $\sigma/\mu$ | $r$ | $CI_{95}$ |
|-----|--------|---|-------|-------|-------|------|---|------|
| H100 | B2/FP16 | 2 | random | 0.4145 | 0.4392 | 1.54%† | 0.9995 | ±0.0044 |
| H100 | B2/FP16 | 2 | scanline | 0.4138 | 0.4355 | 1.59%† | 0.9978 | ±0.0044 |
| H100 | B2/FP16 | 2 | centroid | 0.4148 | 2.4227 | 48.41%† | 1.0002‡ | n/d |
| H100 | B2/FP16 | 2 | Hilbert | 0.4148 | 0.4417 | 1.57%† | 1.0002 | ±0.0044 |
| H100 | B2/FP16 | 2 | K-means CAP | 0.4143 | 0.4411 | 1.81%† | 0.9990 | ±0.0048 |
| H100 | B2/FP16 | 2 | partial DANMP | 0.4156 | 0.4356 | 1.66%† | 1.0022 | ±0.0046 |
| H100 | B2/BF16 | 2 | linear | 0.4140 | 0.4585 | 1.88%† | 1.0000 | — |
| H100 | B2/BF16 | 2 | linear | 0.4239 | 0.4868 | 2.41%† | 1.0000 | — |
| H100 | B2/BF16 | 2 | Morton | 0.4177 | 0.4407 | 1.63%† | 0.9854 | ±0.0056 |
| H100 | B2/BF16 | 2 | Morton | 0.4258 | 0.4687 | 1.90%† | 1.0045 | ±0.0060 |
| H100 | B2/BF16 | 2 | random | 0.4151 | 0.4342 | 1.30%† | 0.9792 | ±0.0053 |
| H100 | B2/BF16 | 2 | random | 0.4274 | 0.4590 | 1.90%† | 1.0083 | ±0.0061 |
| H100 | B2/BF16 | 2 | scanline | 0.4154 | 0.4461 | 1.59%† | 0.9799 | ±0.0055 |
| H100 | B2/BF16 | 2 | scanline | 0.4224 | 0.4480 | 1.54%† | 0.9965 | ±0.0056 |
| H100 | B2/BF16 | 2 | centroid | 0.4126 | 0.4324 | 1.45%† | 0.9733 | ±0.0054 |
| H100 | B2/BF16 | 2 | centroid | 0.4261 | 0.4474 | 1.67%† | 1.0052 | ±0.0058 |
| H100 | B2/BF16 | 2 | Hilbert | 0.4156 | 0.4778 | 2.09%† | 0.9804 | ±0.0061 |
| H100 | B2/BF16 | 2 | Hilbert | 0.4265 | 0.4486 | 1.57%† | 1.0061 | ±0.0057 |
| H100 | B2/BF16 | 2 | K-means CAP | 0.4177 | 0.4389 | 1.65%† | 0.9854 | ±0.0056 |
| H100 | B2/BF16 | 2 | K-means CAP | 0.4256 | 0.4687 | 2.04%† | 1.0040 | ±0.0062 |
| H100 | B2/BF16 | 2 | partial DANMP | 0.4165 | 0.4519 | 1.90%† | 0.9825 | ±0.0059 |
| H100 | B2/BF16 | 2 | partial DANMP | 0.4235 | 0.4532 | 1.91%† | 0.9991 | ±0.0060 |
| H100 | B4/FP32 | 4 | linear | 0.4316 | 0.4553 | 1.69%† | 1.0000 | — |
| H100 | B4/FP32 | 4 | Morton | 0.4315 | 0.4482 | 1.44%† | 0.9998 | ±0.0043 |
| H100 | B4/FP32 | 4 | random | 0.4317 | 0.4533 | 1.48%† | 1.0002 | ±0.0044 |
| H100 | B4/FP32 | 4 | scanline | 0.4334 | 0.5594 | 3.28%† | 1.0042 | ±0.0073 |
| H100 | B4/FP32 | 4 | centroid | 0.4305 | 0.4516 | 1.63%† | 0.9975 | ±0.0046 |
| H100 | B4/FP32 | 4 | Hilbert | 0.4315 | 0.4509 | 1.55%† | 0.9998 | ±0.0045 |
| H100 | B4/FP32 | 4 | K-means CAP | 0.4338 | 0.4513 | 1.48%† | 1.0051 | ±0.0044 |
| H100 | B4/FP32 | 4 | partial DANMP | 0.4333 | 0.4498 | 1.45%† | 1.0039 | ±0.0044 |
| H100 | B4/FP16 | 4 | linear | 0.4166 | 0.8085 | 9.70%† | 1.0000 | — |
| H100 | B4/FP16 | 4 | Morton | 0.4170 | 0.4510 | 1.61%† | 1.0010 | ±0.0193 |
| H100 | B4/FP16 | 4 | random | 0.4134 | 0.4338 | 1.74%† | 0.9923 | ±0.0192 |
| H100 | B4/FP16 | 4 | scanline | 0.4153 | 0.4410 | 1.54%† | 0.9969 | ±0.0192 |
| H100 | B4/FP16 | 4 | centroid | 0.4148 | 0.4447 | 1.49%† | 0.9957 | ±0.0191 |
| H100 | B4/FP16 | 4 | Hilbert | 0.4144 | 2.5922 | 52.56%† | 0.9947‡ | n/d |
| H100 | B4/FP16 | 4 | K-means CAP | 0.4150 | 0.4472 | 1.66%† | 0.9962 | ±0.0192 |
| H100 | B4/FP16 | 4 | partial DANMP | 0.4171 | 0.4549 | 1.87%† | 1.0012 | ±0.0194 |
| H100 | B4/BF16 | 4 | linear | 0.4177 | 0.4478 | 1.75%† | 1.0000 | — |
| H100 | B4/BF16 | 4 | Morton | 0.4185 | 0.4358 | 1.39%† | 1.0019 | ±0.0044 |
| H100 | B4/BF16 | 4 | random | 0.4172 | 0.4462 | 1.49%† | 0.9988 | ±0.0045 |
| H100 | B4/BF16 | 4 | scanline | 0.4172 | 0.4401 | 1.39%† | 0.9988 | ±0.0044 |
| H100 | B4/BF16 | 4 | centroid | 0.4173 | 0.4496 | 1.61%† | 0.9990 | ±0.0046 |
| H100 | B4/BF16 | 4 | Hilbert | 0.4204 | 0.4495 | 1.55%† | 1.0065 | ±0.0046 |
| H100 | B4/BF16 | 4 | K-means CAP | 0.4176 | 0.4421 | 1.41%† | 0.9998 | ±0.0044 |
| H100 | B4/BF16 | 4 | partial DANMP | 0.4205 | 0.4531 | 1.69%† | 1.0067 | ±0.0048 |
| H100 | B8/FP32 | 8 | linear | 0.6690 | 0.7258 | 0.87%† | 1.0000 | — |
| H100 | B8/FP32 | 8 | Morton | 0.6692 | 0.7270 | 0.90%† | 1.0003 | ±0.0024 |
| H100 | B8/FP32 | 8 | random | 0.6692 | 0.7285 | 0.91%† | 1.0003 | ±0.0025 |
| H100 | B8/FP32 | 8 | scanline | 0.6689 | 0.7264 | 0.88%† | 0.9999 | ±0.0024 |
| H100 | B8/FP32 | 8 | centroid | 0.6695 | 0.7275 | 0.90%† | 1.0007 | ±0.0024 |
| H100 | B8/FP32 | 8 | Hilbert | 0.6697 | 0.7281 | 0.90%† | 1.0010 | ±0.0024 |
| H100 | B8/FP32 | 8 | K-means CAP | 0.6696 | 0.7254 | 0.93%† | 1.0009 | ±0.0025 |
| H100 | B8/FP32 | 8 | partial DANMP | 0.6693 | 0.7235 | 0.85%† | 1.0004 | ±0.0024 |
| H100 | B8/FP16 | 8 | linear | 0.4128 | 0.4419 | 1.74%† | 1.0000 | — |
| H100 | B8/FP16 | 8 | Morton | 0.4150 | 0.4373 | 1.64%† | 1.0053 | ±0.0047 |
| H100 | B8/FP16 | 8 | random | 0.4112 | 0.4309 | 1.46%† | 0.9961 | ±0.0044 |
| H100 | B8/FP16 | 8 | scanline | 0.4137 | 0.4376 | 1.47%† | 1.0022 | ±0.0045 |
| H100 | B8/FP16 | 8 | centroid | 0.4131 | 0.4388 | 1.43%† | 1.0007 | ±0.0044 |
| H100 | B8/FP16 | 8 | Hilbert | 0.4129 | 0.4477 | 1.67%† | 1.0002 | ±0.0047 |
| H100 | B8/FP16 | 8 | K-means CAP | 0.4145 | 0.4406 | 1.59%† | 1.0041 | ±0.0046 |
| H100 | B8/FP16 | 8 | partial DANMP | 0.4141 | 0.4481 | 1.69%† | 1.0031 | ±0.0048 |
| H100 | B8/BF16 | 8 | linear | 0.4135 | 0.4465 | 1.60%† | 1.0000 | — |
| H100 | B8/BF16 | 8 | Morton | 0.4133 | 0.4374 | 1.62%† | 0.9995 | ±0.0045 |

*(continued on next page)*

Table 12: Dispatch-order null at $K = 8$. Worst-case $p_{50}$ deviation from the linear baseline across all seven reorderings, per (GPU, configuration, precision) cell. $*$ marks the two batch-1 cells (the smallest, noisiest configuration) that exceed the $\pm 2\%$ band of Section 3.1. Every other cell stays within it.

| GPU | Config/scale | Precision | Max $|r-1|$ |
|---|---|---|---|
| A100 | B1/dec | BF16 | 0.47% |
| A100 | B1/dec | FP16 | 4.95% $*$ |
| A100 | B1/dec | FP32 | 0.79% |
| A100 | B2/enc | BF16 | 0.36% |
| A100 | B2/dec | BF16 | 0.40% |
| A100 | B2/dec | FP16 | 0.58% |
| A100 | B2/dec | FP32 | 0.97% |
| A100 | B4/dec | BF16 | 0.38% |
| A100 | B4/dec | FP16 | 0.38% |
| A100 | B4/dec | FP32 | 0.36% |
| A100 | B8/dec | BF16 | 0.19% |
| A100 | B8/dec | FP16 | 0.36% |
| A100 | B8/dec | FP32 | 0.11% |
| H100 | B1/dec | BF16 | 2.10% $*$ |
| H100 | B1/dec | FP16 | 0.59% |
| H100 | B1/dec | FP32 | 1.23% |
| H100 | B2/enc | BF16 | 0.23% |
| H100 | B2/dec | BF16 | 1.22% |
| H100 | B2/dec | FP16 | 1.20% |
| H100 | B2/dec | FP32 | 0.81% |
| H100 | B4/dec | BF16 | 0.73% |
| H100 | B4/dec | FP16 | 0.47% |
| H100 | B4/dec | FP32 | 0.45% |
| H100 | B8/dec | BF16 | 0.94% |
| H100 | B8/dec | FP16 | 1.04% |
| H100 | B8/dec | FP32 | 0.42% |

*(Table 11 continued from previous page.)*

| GPU | Config | B | Order | $p_{50}$ (ms) | $p_{99}$ (ms) | $\sigma/\mu$ | $r$ | CI$_{95}$ |
|---|---|---|---|---|---|---|---|---|
| H100 | B8/BF16 | 8 | random | 0.4141 | 0.4318 | 1.47%† | 1.0015 | ±0.0043 |
| H100 | B8/BF16 | 8 | scanline | 0.4153 | 0.4409 | 1.52%† | 1.0044 | ±0.0043 |
| H100 | B8/BF16 | 8 | centroid | 0.4130 | 0.4358 | 1.60%† | 0.9988 | ±0.0044 |
| H100 | B8/BF16 | 8 | Hilbert | 0.4159 | 0.4360 | 1.37%† | 1.0058 | ±0.0041 |
| H100 | B8/BF16 | 8 | K-means CAP | 0.4160 | 0.4518 | 1.66%† | 1.0060 | ±0.0045 |
| H100 | B8/BF16 | 8 | partial DANMP | 0.4157 | 0.4603 | 1.83%† | 1.0053 | ±0.0048 |

Table 14: Dispatch-order null across level counts $L \in \{2, 3, 5\}$. Worst-case $p_{50}$ deviation from the linear baseline across all seven reorderings, per (GPU, $L$, configuration, precision) cell, grouped by $L$. Each group repeats the main $L=4$ comparison (Section 3.1). Decoder-scale configurations (800×1333) are listed before the large-resolution encoder-scale (1536×2048) cells, which form a separate operating point (Section 3.1). $*$ marks a jitter outlier (A100, $L=2$, batch-8, FP16), where run-to-run variance doubles.

| $L$ | GPU | Config/scale | Precision | Max $|r-1|$ |
|---|---|---|---|---|
| $L = 2$ | A100 | B1/dec | BF16 | 0.39% |
| $L = 2$ | A100 | B1/dec | FP16 | 0.91% |
| $L = 2$ | A100 | B1/dec | FP32 | 1.17% |

*(Table 14 continued from previous page.)*

| L | GPU | Config/scale | Precision | Max $|r-1|$ |
|---|-----|--------------|-----------|-------------|
| $L=2$ | A100 | B2/dec | BF16 | 0.59% |
| $L=2$ | A100 | B2/dec | FP16 | 0.27% |
| $L=2$ | A100 | B2/dec | FP32 | 0.58% |
| $L=2$ | A100 | B4/dec | BF16 | 0.58% |
| $L=2$ | A100 | B4/dec | FP16 | 0.30% |
| $L=2$ | A100 | B4/dec | FP32 | 0.28% |
| $L=2$ | A100 | B8/dec | BF16 | 0.79% |
| $L=2$ | A100 | B8/dec | FP16 | 6.61% * |
| $L=2$ | A100 | B8/dec | FP32 | 0.33% |
| $L=2$ | H100 | B1/dec | BF16 | 0.43% |
| $L=2$ | H100 | B1/dec | FP16 | 0.46% |
| $L=2$ | H100 | B1/dec | FP32 | 0.72% |
| $L=2$ | H100 | B2/dec | BF16 | 0.77% |
| $L=2$ | H100 | B2/dec | FP16 | 0.69% |
| $L=2$ | H100 | B2/dec | FP32 | 0.85% |
| $L=2$ | H100 | B4/dec | BF16 | 0.57% |
| $L=2$ | H100 | B4/dec | FP16 | 0.50% |
| $L=2$ | H100 | B4/dec | FP32 | 0.42% |
| $L=2$ | H100 | B8/dec | BF16 | 0.47% |
| $L=2$ | H100 | B8/dec | FP16 | 0.64% |
| $L=2$ | H100 | B8/dec | FP32 | 0.31% |
| $L=3$ | A100 | B1/dec | BF16 | 0.19% |
| $L=3$ | A100 | B1/dec | FP16 | 0.94% |
| $L=3$ | A100 | B1/dec | FP32 | 0.58% |
| $L=3$ | A100 | B2/dec | BF16 | 0.40% |
| $L=3$ | A100 | B2/dec | FP16 | 0.09% |
| $L=3$ | A100 | B2/dec | FP32 | 0.22% |
| $L=3$ | A100 | B4/dec | BF16 | 0.39% |
| $L=3$ | A100 | B4/dec | FP16 | 0.21% |
| $L=3$ | A100 | B4/dec | FP32 | 0.38% |
| $L=3$ | A100 | B8/dec | BF16 | 0.40% |
| $L=3$ | A100 | B8/dec | FP16 | 0.38% |
| $L=3$ | A100 | B8/dec | FP32 | 0.00% |
| $L=3$ | H100 | B1/dec | BF16 | 0.56% |
| $L=3$ | H100 | B1/dec | FP16 | 1.80% |
| $L=3$ | H100 | B1/dec | FP32 | 1.05% |
| $L=3$ | H100 | B2/dec | BF16 | 1.07% |
| $L=3$ | H100 | B2/dec | FP16 | 1.06% |
| $L=3$ | H100 | B2/dec | FP32 | 0.33% |
| $L=3$ | H100 | B4/dec | BF16 | 0.87% |
| $L=3$ | H100 | B4/dec | FP16 | 0.63% |
| $L=3$ | H100 | B4/dec | FP32 | 0.16% |
| $L=3$ | H100 | B8/dec | BF16 | 0.42% |
| $L=3$ | H100 | B8/dec | FP16 | 0.66% |
| $L=3$ | H100 | B8/dec | FP32 | 0.20% |
| $L=5$ | A100 | B1/dec | BF16 | 0.58% |
| $L=5$ | A100 | B1/dec | FP16 | 0.45% |
| $L=5$ | A100 | B1/dec | FP32 | 0.39% |
| $L=5$ | A100 | B2/dec | BF16 | 0.40% |
| $L=5$ | A100 | B2/dec | FP16 | 0.56% |
| $L=5$ | A100 | B2/dec | FP32 | 0.29% |
| $L=5$ | A100 | B4/dec | BF16 | 0.19% |
| $L=5$ | A100 | B4/dec | FP16 | 0.67% |
| $L=5$ | A100 | B4/dec | FP32 | 0.04% |

*(Table 14 continued from previous page.)*

| L | GPU | Config/scale | Precision | Max $|r-1|$ |
|---|---|---|---|---|
| $L=5$ | A100 | B8/dec | BF16 | 0.80% |
| $L=5$ | A100 | B8/dec | FP16 | 1.32% |
| $L=5$ | A100 | B8/dec | FP32 | 0.02% |
| $L=5$ | H100 | B1/dec | BF16 | 0.95% |
| $L=5$ | H100 | B1/dec | FP16 | 1.86% |
| $L=5$ | H100 | B1/dec | FP32 | 0.94% |
| $L=5$ | H100 | B2/dec | BF16 | 0.98% |
| $L=5$ | H100 | B2/dec | FP16 | 0.59% |
| $L=5$ | H100 | B2/dec | FP32 | 0.28% |
| $L=5$ | H100 | B4/dec | BF16 | 0.59% |
| $L=5$ | H100 | B4/dec | FP16 | 0.75% |
| $L=5$ | H100 | B4/dec | FP32 | 0.19% |
| $L=5$ | H100 | B8/dec | BF16 | 0.21% |
| $L=5$ | H100 | B8/dec | FP16 | 0.25% |
| $L=5$ | H100 | B8/dec | FP32 | 1.33% |

Table 17: Shipped-library MSDA operators and FlashAttention at the encoder operating points (BF16, $Q$ = encoder tokens; CS = Cityscapes). Rows are the MSDA implementations distributed in the PyTorch ecosystem, Transformers (Wolf et al., 2020) (native-PyTorch bilinear-sampling and a compiled CUDA kernel) and mmcv/mmdetection (MMCV Contributors, 2018), plus full $O(N^2)$ self-attention via FlashAttention (a *different* operator over the same $N$ tokens).

| GPU/Point | Implementation | Fwd (ms) | Bwd (ms) | Peak (MB) |
|---|---|---|---|---|
| A100 DETR/COCO ($N$=22015) | Transformers (Wolf et al., 2020) (PyTorch) | 8.47 | 432.68 | 1302 |
| | Transformers (Wolf et al., 2020) (CUDA) | 2.09 | 12.60 | 388 |
| | CUDA (Zhu et al., 2021) | 1.93 | 5.92 | 496 |
| | CUDA *ours* | 0.56 | 4.85 | 474 |
| | Triton, query-block | 0.59 | 120.89 | 388 |
| | FlashAttention (full attn.) | 7.99 | 21.44 | 519 |
| A100 DETR/CS ($N$=43520) | Transformers (Wolf et al., 2020) (PyTorch) | 17.10 | 663.84 | 2571 |
| | Transformers (Wolf et al., 2020) (CUDA) | 4.09 | 25.32 | 765 |
| | CUDA (Zhu et al., 2021) | 3.99 | 12.87 | 980 |
| | CUDA *ours* | 1.27 | 9.79 | 935 |
| | Triton, query-block | 0.97 | 151.87 | 765 |
| | FlashAttention (full attn.) | 30.60 | 72.74 | 1023 |
| A100 M2F/COCO ($N$=21504) | Transformers (Wolf et al., 2020) (PyTorch) | 5.47 | 155.05 | 991 |
| | Transformers (Wolf et al., 2020) (CUDA) | 1.60 | 9.01 | 362 |
| | CUDA (Zhu et al., 2021) | 1.53 | 4.66 | 452 |
| | CUDA *ours* | 0.44 | 3.79 | 415 |
| | Triton, query-block | 0.50 | 40.75 | 362 |
| | FlashAttention (full attn.) | 7.49 | 18.03 | 490 |
| A100 M2F/CS ($N$=43008) | Transformers (Wolf et al., 2020) (PyTorch) | 11.37 | 232.88 | 1980 |
| | Transformers (Wolf et al., 2020) (CUDA) | 3.16 | 19.04 | 724 |
| | CUDA (Zhu et al., 2021) | 3.24 | 10.31 | 903 |
| | CUDA *ours* | 0.90 | 7.78 | 830 |
| | Triton, query-block | 0.77 | 58.32 | 724 |
| | FlashAttention (full attn.) | 29.71 | 70.51 | 979 |
| H100 DETR/COCO ($N$=22015) | Transformers (Wolf et al., 2020) (PyTorch) | 5.42 | 15.14 | 1302 |
| | Transformers (Wolf et al., 2020) (CUDA) | 1.41 | 7.12 | 388 |
| | CUDA (Zhu et al., 2021) | 1.16 | 2.87 | 496 |

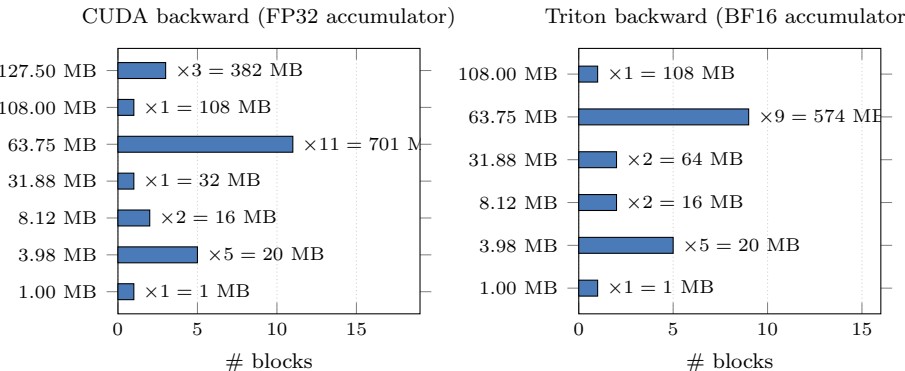

Figure 6: Per-allocation block-size histogram during the encoder-scale backward pass at BF16, captured on the A100 partition. Blocks $\geq$ 1 MB are shown; sub-MB scaffolding is elided. The CUDA path's top three 127.50/63.75/31.88 MB FP32 scratch buffers (the gradient-of-value, gradient-of-sampling-location, and gradient-of-attention accumulators, allocated in single precision regardless of the input dtype) are absent from the Triton path. Sizes are reported with the binary mebibyte convention ($1\,\text{MB} = 2^{20}\,\text{B}$) for consistency with PyTorch's peak-allocation reporting.

| GPU/Point | Implementation | Fwd (ms) | Bwd (ms) | Peak (MB) |
|---|---|---|---|---|
| | CUDA *ours* | 0.36 | 2.85 | 474 |
| | Triton, query-block | 0.33 | 1.10 | 388 |
| | FlashAttention (full attn.) | 4.48 | 11.01 | 519 |
| H100 DETR/CS ($N$=43520) | Transformers (Wolf et al., 2020) (PyTorch) | 10.92 | 28.47 | 2573 |
| | Transformers (Wolf et al., 2020) (CUDA) | 2.70 | 14.01 | 766 |
| | CUDA (Zhu et al., 2021) | 2.26 | 6.03 | 981 |
| | CUDA *ours* | 0.69 | 5.75 | 936 |
| | Triton, query-block | 0.53 | 2.13 | 766 |
| | FlashAttention (full attn.) | 15.52 | 38.81 | 1024 |
| H100 M2F/COCO ($N$=21504) | Transformers (Wolf et al., 2020) (PyTorch) | 3.43 | 10.16 | 991 |
| | Transformers (Wolf et al., 2020) (CUDA) | 1.08 | 5.29 | 363 |
| | CUDA (Zhu et al., 2021) | 0.89 | 2.18 | 452 |
| | CUDA *ours* | 0.30 | 2.18 | 415 |
| | Triton, query-block | 0.31 | 0.85 | 363 |
| | FlashAttention (full attn.) | 3.89 | 9.81 | 491 |
| H100 M2F/CS ($N$=43008) | Transformers (Wolf et al., 2020) (PyTorch) | 7.07 | 19.15 | 1980 |
| | Transformers (Wolf et al., 2020) (CUDA) | 2.07 | 10.51 | 725 |
| | CUDA (Zhu et al., 2021) | 1.83 | 4.89 | 903 |
| | CUDA *ours* | 0.51 | 4.60 | 830 |
| | Triton, query-block | 0.45 | 1.68 | 725 |
| | FlashAttention (full attn.) | 15.04 | 37.54 | 979 |

Table 18 projects the measured full-model training-step time (forward + backward + AdamW update, BF16) over a COCO train2017 epoch, giving practitioners a direct wall-clock sense of impact.

Figure 6 shows the block-size histogram of the per-allocation memory snapshot at the encoder cell, comparing the CUDA backend against our query-block Triton backend during the backward pass. The asymmetry is dominated by three FP32 scratch buffers in the CUDA path (the topmost 127.50/63.75/31.88 MB blocks) that are absent from the Triton path; this is the FP32-accumulator cost the design choice removes (Section 3.3).

Table 13: Dispatch-order null at $K = 16$. Worst-case $p_{50}$ deviation from the linear baseline across all seven reorderings, per (GPU, configuration, precision) cell. Every cell stays within the $\pm 2\%$ band of Section 3.1.

| GPU | Config/scale ($K = 16$) | Precision | Max $|r - 1|$ |
|-----|-------------------------|-----------|---------------|
| A100 | B1/dec | BF16 | 0.38% |
| A100 | B1/dec | FP16 | 0.57% |
| A100 | B1/dec | FP32 | 0.69% |
| A100 | B2/enc | BF16 | 0.25% |
| A100 | B2/dec | BF16 | 0.79% |
| A100 | B2/dec | FP16 | 0.59% |
| A100 | B2/dec | FP32 | 0.69% |
| A100 | B4/dec | BF16 | 0.38% |
| A100 | B4/dec | FP16 | 0.48% |
| A100 | B4/dec | FP32 | 0.37% |
| A100 | B8/dec | BF16 | 0.39% |
| A100 | B8/dec | FP16 | 0.36% |
| A100 | B8/dec | FP32 | 0.12% |
| H100 | B1/dec | BF16 | 0.61% |
| H100 | B1/dec | FP16 | 0.61% |
| H100 | B1/dec | FP32 | 0.84% |
| H100 | B2/enc | BF16 | 0.11% |
| H100 | B2/dec | BF16 | 0.92% |
| H100 | B2/dec | FP16 | 1.43% |
| H100 | B2/dec | FP32 | 1.58% |
| H100 | B4/dec | BF16 | 0.38% |
| H100 | B4/dec | FP16 | 0.89% |
| H100 | B4/dec | FP32 | 0.54% |
| H100 | B8/dec | BF16 | 0.40% |
| H100 | B8/dec | FP16 | 0.64% |
| H100 | B8/dec | FP32 | 0.83% |

Table 19 reports median ($p_{50}$), tail ($p_{99}$), and standard deviation ($\sigma$) for the forward and backward passes of every backend at the decoder configuration on both GPUs. All five backends exhibit sub-percent relative standard deviations, confirming that the $p_{50}$ values in Table 5 are stable.

Tables 20 and 21 support the generalization analysis of Section 4.2.

Tables 22 and 23 list the complete operator-level sweep across all twelve configurations, three precisions, and five backends, one table per GPU. One anomalous cell: the Triton point-parallel backward on H100 at B4_800×1333_enc FP16 (126.6 ms) is 2.5× slower than the same kernel at BF16 (51.0 ms) and 5.5× slower than FP32 (23.0 ms). This inversion affects only the point-parallel tiling, which is already identified as a throughput failure in Section 3.2; the recommended query-block tiling shows the expected precision ordering at the same configuration (3.1 ms FP16 vs. 3.1 ms BF16). The anomaly is specific to the point-parallel tiling at FP16 on the Triton version shipped with PyTorch 2.9.0 (pinned in Section 5); the recommended query-block tiling shows the expected precision ordering at the same configuration and is unaffected. Decoder configurations correspond to the standard Deformable DETR decoder stage; encoder configurations (suffixed "enc") correspond to encoder-stage feature maps of roughly the resolutions used by Deformable DETR, DINO, and Mask2Former. All latencies are $p_{50}$ in milliseconds over 100 measurement iterations after 10 warmup iterations.

# B  Numerical Parity

Table 24 reports maximum absolute error and mean relative error of each backend against the FP32 reference implementation (cast to float64 before comparison). At FP32 the custom backends match the reference to $1.6 \cdot 10^{-6}$, the magnitude expected from reordered parallel summation of single-precision floats. At FP16 and BF16 the custom backends track the reference within 0.2 % of its own precision-bounded error; the mean

Table 15: Isolated MSDA-layer speedup at the true model operating points (BF16, $Q$ = encoder tokens; CS = Cityscapes). DeformableDETR uses $L$=4 levels, Mask2Former $L$=3. Columns are forward and backward speedup over eager execution (*eag*) and the CUDA kernel of Zhu et al. (2021) (*ref*). Graph compilation stays at $\leq 1\times$ vs eager.

| Point ($Q$) | GPU | Kernel | Fwd $\times$ | | Bwd $\times$ | |
| | | | eag | ref | eag | ref |
|---|---|---|---|---|---|---|
| DETR/COCO (22015) | A100 | graph-comp. | 0.97 | – | 0.99 | – |
| | | CUDA *ours* | 9.80 | 1.46 | 50.86 | 1.09 |
| | | Triton, query-block | 10.30 | 1.54 | 3.02 | 0.06 |
| | | FP32-accum. variant | 10.52 | 1.57 | 55.08 | 1.18 |
| DETR/COCO (22015) | H100 | graph-comp. | 0.95 | – | 1.01 | – |
| | | CUDA *ours* | 9.23 | 1.29 | 2.89 | 0.95 |
| | | Triton, query-block | 9.55 | 1.33 | 8.07 | 2.65 |
| | | FP32-accum. variant | 9.53 | 1.33 | 6.98 | 2.29 |
| DETR/CS (43520) | A100 | graph-comp. | 0.98 | – | 1.01 | – |
| | | CUDA *ours* | 8.86 | 1.46 | 40.46 | 1.16 |
| | | Triton, query-block | 11.28 | 1.86 | 3.83 | 0.11 |
| | | FP32-accum. variant | 11.45 | 1.89 | 45.26 | 1.29 |
| DETR/CS (43520) | H100 | graph-comp. | 0.97 | – | 1.02 | – |
| | | CUDA *ours* | 10.61 | 1.53 | 2.69 | 0.99 |
| | | Triton, query-block | 13.13 | 1.89 | 8.19 | 3.02 |
| | | FP32-accum. variant | 13.12 | 1.89 | 6.95 | 2.56 |
| M2F/COCO (21504) | A100 | graph-comp. | 0.91 | – | 0.98 | – |
| | | CUDA *ours* | 6.73 | 1.20 | 24.25 | 1.11 |
| | | Triton, query-block | 5.42 | 0.97 | 3.37 | 0.15 |
| | | FP32-accum. variant | 6.78 | 1.21 | 26.13 | 1.20 |
| M2F/COCO (21504) | H100 | graph-comp. | 0.92 | – | 1.03 | – |
| | | CUDA *ours* | 6.40 | 1.10 | 2.47 | 0.96 |
| | | Triton, query-block | 6.47 | 1.11 | 6.30 | 2.44 |
| | | FP32-accum. variant | 6.40 | 1.10 | 5.47 | 2.11 |
| M2F/CS (43008) | A100 | graph-comp. | 0.96 | – | 1.00 | – |
| | | CUDA *ours* | 7.46 | 1.46 | 18.33 | 1.18 |
| | | Triton, query-block | 8.29 | 1.62 | 3.48 | 0.22 |
| | | FP32-accum. variant | 8.79 | 1.72 | 20.11 | 1.29 |
| M2F/CS (43008) | H100 | graph-comp. | 0.96 | – | 1.03 | – |
| | | CUDA *ours* | 8.57 | 1.37 | 2.33 | 1.01 |
| | | Triton, query-block | 10.01 | 1.61 | 6.57 | 2.85 |
| | | FP32-accum. variant | 10.01 | 1.61 | 5.62 | 2.44 |

relative error exceeding 1 at BF16 reflects BF16's 7-bit mantissa blowup on small-magnitude values under uniform random inputs rather than any kernel defect.

## C Full Scatter-Add Sweep

We sweep PyTorch's standard scatter-add operator across $(M, D, N) \in \{32{,}768, 131{,}072\} \times \{64, 256, 1024\} \times \{16{,}384, 65{,}536, 262{,}144\}$, over uniform and Zipf-$s$=1.5 index distributions, at BF16, FP16, and FP32 precision, on A100 and H100. The Zipf distribution produces single-address cross-warp contention of up to $\sim 10^5$ writes to the worst row. Table 20 reports six representative cells at $D$=256; the full 72-cell matrix will accompany the source release upon acceptance. At low contention FP16 is 20 %–30 % faster than BF16 on A100, consistent with the native-vs-CAS expectation, but at the worst Zipf cell FP16 runs 1.5–2.1$\times$ slower than BF16 on both GPUs: the native FP16 atomic hardware scales worse than the software CAS loop

Table 16: Forward DRAM-bandwidth utilization (% of peak) and L2 hit rate at the operating points (BF16, NSight Compute; CS = Cityscapes). The CUDA (Zhu et al., 2021), CUDA *ours*, and Triton column groups report each backend's utilization (Section 5).

| Point | GPU | CUDA (Zhu et al., 2021) | | CUDA *ours* | | Triton | |
|---|---|---|---|---|---|---|---|
| | | BW% | L2% | BW% | L2% | BW% | L2% |
| DETR/COCO | A100 | 45 | 81 | 14 | 91 | 14 | 92 |
| DETR/COCO | H100 | 66 | 85 | 18 | 93 | 24 | 91 |
| DETR/CS | A100 | 59 | 73 | 35 | 77 | 15 | 93 |
| DETR/CS | H100 | 92 | 76 | 40 | 84 | 26 | 92 |
| M2F/COCO | A100 | 32 | 84 | 10 | 91 | 15 | 92 |
| M2F/COCO | H100 | 46 | 90 | 16 | 91 | 28 | 91 |
| M2F/CS | A100 | 59 | 69 | 16 | 83 | 18 | 92 |
| M2F/CS | H100 | 94 | 74 | 20 | 90 | 31 | 92 |

Table 18: Per-epoch training wallclock at the models' operating points (BF16, pretrained weights, AdamW). *step* is a measured training step (forward + backward + optimizer update); *h/epoch* projects it over a COCO train2017 epoch (118287 images); $\Delta$ is the per-epoch reduction over the CUDA kernel of Zhu et al. (2021). † marks the A100 Triton step, where the BF16 backward hits the atomic cliff (Section 3.3). CUDA *ours* or the FP32-accumulator variant is the A100 choice.

| Model | GPU | Kernel | step (ms) | h/epoch | $\Delta$ |
|---|---|---|---|---|---|
| Deformable-DETR | A100 | CUDA (Zhu et al., 2021) | 115.7 | 1.90 | − |
| | | CUDA *ours* | 105.5 | 1.73 | −9% |
| | | Triton | 738.6 | 12.13 | +538%[†] |
| Deformable-DETR | H100 | CUDA (Zhu et al., 2021) | 74.4 | 1.22 | − |
| | | CUDA *ours* | 69.1 | 1.14 | −7% |
| | | Triton | 59.5 | 0.98 | −20% |
| Mask2Former | A100 | CUDA (Zhu et al., 2021) | 167.6 | 2.75 | − |
| | | CUDA *ours* | 159.2 | 2.62 | −5% |
| | | Triton | 416.8 | 6.85 | +149%[†] |
| Mask2Former | H100 | CUDA (Zhu et al., 2021) | 89.5 | 1.47 | − |
| | | CUDA *ours* | 86.2 | 1.42 | −4% |
| | | Triton | 78.1 | 1.28 | −13% |

under extreme contention. Between-run spread reaches ∼55 % on A100 BF16 at the worst cell, intrinsic to CAS-retry contention rather than thermal drift (FP32 at the same cell is stable to 1 %). Table 21 reports the GPU hardware counters at the worst-contention cell for both operators; the atomic-lowering *flips* between an atomic-add and a hardware-reduction instruction family across SM 8.0/9.0 for MSDA's Triton-compiled backward but not for the nvcc-compiled scatter-add operator.

## D   Technique Ablation

Table 25 isolates the contribution of each kernel-design technique at the encoder-scale configuration ($B=2$, 1536×2048, $L=4$, BF16), where the differences are largest; at decoder scale all custom backends converge within 10 % of each other. The CUDA and Triton paths are independent tracks; the Triton rows are not cumulative with the CUDA rows.

Read-side optimizations account for the majority of the CUDA forward improvement on A100 (6.65 → 3.12 ms, 2.1×); write-side relaxed atomics add only 7 % more (3.12 → 2.92 ms), confirming that the forward bottleneck is read bandwidth, not atomic contention. The tiling strategy is the single largest Triton lever (8.2× on

Table 19: Backward and forward latency dispersion: median ($p_{50}$), tail ($p_{99}$), and standard deviation ($\sigma$) per backend on A100 and H100. Config: B4 800×1333, BF16, eager, external softmax; $N$=100 samples.

| | | A100 | | | H100 | | |
| | | $p_{50}$ (ms) | $p_{99}$ (ms) | $\sigma$ (ms) | $p_{50}$ (ms) | $p_{99}$ (ms) | $\sigma$ (ms) |
| Backend | Phase | | | | | | |
|---|---|---|---|---|---|---|---|
| Reference | Fwd | 1.416 | 2.681 | 0.127 | 1.053 | 2.193 | 0.114 |
| | Bwd | 5.440 | 6.656 | 0.163 | 2.142 | 3.274 | 0.143 |
| CUDA (Zhu et al., 2021) | Fwd | 0.544 | 0.567 | 0.005 | 0.446 | 0.474 | 0.007 |
| | Bwd | 1.495 | 2.793 | 0.130 | 1.170 | 2.382 | 0.121 |
| CUDA *ours* | Fwd | 0.564 | 0.607 | 0.008 | 0.472 | 0.504 | 0.007 |
| | Bwd | 1.522 | 2.788 | 0.127 | 1.206 | 2.612 | 0.141 |
| Triton, point-parallel | Fwd | 0.561 | 0.632 | 0.010 | 0.441 | 0.473 | 0.006 |
| | Bwd | 1.777 | 2.351 | 0.058 | 1.160 | 2.353 | 0.124 |
| Triton, query-block | Fwd | 0.534 | 0.587 | 0.008 | 0.445 | 0.481 | 0.007 |
| | Bwd | 1.572 | 2.297 | 0.074 | 1.187 | 2.414 | 0.124 |

Table 20: Scatter-add $p_{50}$ latency at six representative cells ($M$=32,768, $D$=256, $N \in \{16k, 64k, 256k\}$, uniform or Zipf-$s$=1.5 index distribution) on A100 and H100. FP32 stays fast across all collision rates; BF16 collapses two to three orders of magnitude under Zipf contention on both architectures. Full 72-cell sweep in the source CSVs.

| $N$ | Dist. | worst-row coll. | A100 BF16 (ms) | A100 FP32 (ms) | H100 BF16 (ms) | H100 FP32 (ms) |
|---|---|---|---|---|---|---|
| 16k | uniform | 5 | 0.109 | 0.0799 | 0.0395 | 0.0477 |
| 16k | zipf | 6251 | 130.6 | 0.114 | 72.3 | 0.0824 |
| 64k | uniform | 10 | 0.366 | 0.254 | 0.125 | 0.151 |
| 64k | zipf | 25,174 | 249.4 | 0.386 | 369.3 | 0.285 |
| 256k | uniform | 23 | 1.42 | 0.954 | 0.464 | 0.561 |
| 256k | zipf | 100,797 | 718.7 | 1.48 | 1074.5 | 1.09 |

the A100 forward when switching from point-parallel to query-block). The backward pass does not benefit from kernel-design techniques on A100 at BF16: the CUDA backward is unchanged between the read-side and write-side rows (16.86 vs 17.21 ms) and the Triton query-block backward (123.4 ms) is *slower* than the point-parallel backward (94.3 ms) because the former emits one relaxed BF16 atomic per corner where the latter amortizes atomics across a larger tile. Both observations reflect the hardware-gated CAS loop of Section 3.3. On H100, where hardware BF16 atomics resolve the contention, the Triton query-block backward drops to 4.3 ms, confirming the residual is architectural rather than algorithmic.

# E  Kernel Pseudocode

This appendix gives pseudocode for the two kernel patterns that are load-bearing for the speedups reported in the body: warp-level sharing of sampling coordinates in the CUDA forward path, and relaxed-ordering scattered atomic accumulation in the Triton backward path. Both listings elide error checks, boundary masking, and template parameters for readability; the full sources will accompany the release upon acceptance.

# F  Partial DANMP Port: Sampling-Point Clustering Ablation

The CAP analogue cited in Section 3.1 deviates from DANMP's full recipe on four axes: (i) we cluster the mean cross-level reference points (deterministic per query, nearly uniform in image space), whereas DANMP clusters the sampling points (reference + learned offset, where the hotspot signal lives); (ii) we use fixed-$k$ K-means with $k \approx 2\sqrt{Q}$, whereas DANMP uses a 9×9 pixel-radius DBSCAN-style variable-$k$ hotspot detector; (iii) we cluster all queries, whereas DANMP samples 20% and maps the rest to nearest

Table 21: Kernel-level counter fingerprint for PyTorch's scatter-add operator (nvcc-compiled) and the MSDA backward (Triton-compiled) at BF16, on A100 (SM 8.0) and H100 (SM 9.0). *Kernel-replay profiling underreports wall time by ~1.8–2.2×. Benchmark $p_{50}$ values are used for absolute latencies.

| Counter | Scatter-add | | MSDA bwd | |
|---|---|---|---|---|
| | A100 | H100 | A100 | H100 |
| Kernel time* (ms) | 603.90 | 426.98 | 109.70 | 0.74 |
| L1 atomic-add sectors ($\times 10^6$) | 525 | 492 | 447 | 0 |
| L1 hardware-reduction sectors ($\times 10^6$) | 0 | 0 | 0 | 35 |
| L2 atomic-add sectors ($\times 10^6$) | 633 | 739 | 672 | 0 |
| L2 hardware-reduction sectors ($\times 10^6$) | 0 | 0 | 0 | 52 |
| DRAM traffic (MB) | 148 | 155 | 170 | 162 |

Table 22: Full operator-level latency sweep on A100. $p_{50}$ in milliseconds, eager mode, external softmax.

| Config | Precision | Reference Fwd | Reference Bwd | CUDA (Zhu et al., 2021) Fwd | CUDA (Zhu et al., 2021) Bwd | CUDA *ours* Fwd | CUDA *ours* Bwd | Triton, point-parallel Fwd | Triton, point-parallel Bwd | Triton, query-block Fwd | Triton, query-block Bwd |
|---|---|---|---|---|---|---|---|---|---|---|---|
| | FP32 | 1.127 | 2.530 | 0.504 | 1.348 | 0.598 | 1.454 | 0.554 | 1.484 | 0.564 | 1.519 |
| B1 640 d256 L4 | FP16 | 1.137 | 2.549 | 0.573 | 1.492 | 0.603 | 1.544 | 0.563 | 1.507 | 0.572 | 1.536 |
| | BF16 | 1.102 | 4.204 | 0.546 | 1.445 | 0.576 | 1.494 | 0.537 | 1.469 | 0.547 | 1.498 |
| | FP32 | 1.465 | 3.589 | 0.502 | 1.384 | 0.579 | 1.458 | 0.570 | 1.481 | 0.546 | 1.512 |
| B2 1280 d256 L4 | FP16 | 1.417 | 3.272 | 0.551 | 1.515 | 0.563 | 1.519 | 0.558 | 1.564 | 0.532 | 1.500 |
| | BF16 | 1.417 | 6.617 | 0.551 | 1.516 | 0.565 | 1.526 | 0.560 | 2.420 | 0.536 | 1.776 |
| | FP32 | 1.739 | 5.236 | 0.501 | 1.398 | 0.580 | 1.459 | 0.580 | 1.722 | 0.550 | 1.518 |
| B2 1536x2048 d256 L4 | FP16 | 1.597 | 4.198 | 0.588 | 1.505 | 0.571 | 1.530 | 0.564 | 1.746 | 0.541 | 1.505 |
| | BF16 | 1.601 | 6.796 | 0.585 | 1.514 | 0.570 | 1.530 | 0.568 | 2.160 | 0.539 | 1.741 |
| | FP32 | 30.956 | 80.523 | 6.720 | 24.938 | 5.292 | 19.705 | 15.138 | 51.214 | 3.430 | 20.756 |
| B2 1536x2048 d256 L4 enc | FP16 | 25.606 | 85.265 | 6.687 | 24.320 | 2.989 | 17.274 | 16.301 | 70.987 | 2.135 | 13.103 |
| | BF16 | 25.569 | 692.335 | 6.649 | 24.280 | 2.920 | 17.212 | 16.447 | 94.253 | 1.997 | 123.404 |
| | FP32 | 1.026 | 2.222 | 0.498 | 1.369 | 0.582 | 1.454 | 0.541 | 1.487 | 0.563 | 1.537 |
| B2 800x1333 d256 L2 | FP16 | 1.002 | 2.209 | 0.570 | 1.501 | 0.591 | 1.525 | 0.551 | 1.483 | 0.575 | 1.537 |
| | BF16 | 0.983 | 2.365 | 0.551 | 1.457 | 0.573 | 1.477 | 0.545 | 1.453 | 0.553 | 1.503 |
| | FP32 | 1.323 | 2.841 | 0.501 | 1.371 | 0.583 | 1.458 | 0.542 | 1.482 | 0.550 | 1.510 |
| B2 800x1333 d256 L4 | FP16 | 1.310 | 2.838 | 0.575 | 1.510 | 0.597 | 1.535 | 0.556 | 1.503 | 0.564 | 1.515 |
| | BF16 | 1.293 | 4.368 | 0.560 | 1.468 | 0.578 | 1.496 | 0.537 | 1.458 | 0.549 | 1.486 |
| | FP32 | 10.264 | 30.562 | 2.193 | 7.898 | 1.396 | 6.330 | 5.156 | 18.343 | 1.173 | 7.402 |
| B2 800x1333 d256 L4 enc | FP16 | 8.593 | 32.406 | 2.167 | 7.574 | 0.862 | 5.649 | 5.513 | 26.963 | 0.728 | 4.671 |
| | BF16 | 8.593 | 350.315 | 2.168 | 7.553 | 0.863 | 5.636 | 5.546 | 50.364 | 0.710 | 86.413 |
| | FP32 | 1.951 | 6.730 | 0.490 | 1.359 | 0.575 | 1.452 | 0.536 | 1.544 | 0.544 | 1.499 |
| B2 800x1333 d256 L5 | FP16 | 1.801 | 5.366 | 0.598 | 1.496 | 0.589 | 1.527 | 0.550 | 1.479 | 0.558 | 1.506 |
| | BF16 | 1.783 | 7.389 | 0.583 | 1.456 | 0.571 | 1.486 | 0.532 | 1.522 | 0.542 | 1.853 |
| | FP32 | 1.533 | 3.949 | 0.503 | 1.366 | 0.584 | 1.447 | 0.544 | 1.481 | 0.554 | 1.512 |
| B2 800x1333 d512 L4 | FP16 | 1.458 | 4.047 | 0.570 | 1.504 | 0.591 | 1.530 | 0.551 | 1.483 | 0.560 | 1.519 |
| | BF16 | 1.445 | 6.789 | 0.545 | 1.464 | 0.565 | 1.486 | 0.528 | 1.816 | 0.537 | 2.255 |
| | FP32 | 1.497 | 3.775 | 0.492 | 1.384 | 0.576 | 1.465 | 0.563 | 1.491 | 0.543 | 1.526 |
| B4 800x1333 d256 L4 | FP16 | 1.410 | 3.396 | 0.546 | 1.495 | 0.567 | 1.522 | 0.563 | 1.475 | 0.534 | 1.505 |
| | BF16 | 1.416 | 5.440 | 0.544 | 1.495 | 0.564 | 1.522 | 0.561 | 1.777 | 0.534 | 1.572 |
| | FP32 | 19.982 | 59.279 | 4.312 | 15.288 | 2.724 | 12.164 | 10.244 | 34.546 | 2.277 | 14.294 |
| B4 800x1333 d256 L4 enc | FP16 | 16.674 | 64.131 | 4.284 | 14.881 | 1.701 | 11.054 | 11.009 | 53.709 | 1.436 | 9.140 |
| | BF16 | 16.616 | 686.856 | 4.263 | 14.819 | 1.673 | 11.001 | 11.195 | 100.071 | 1.367 | 167.570 |
| | FP32 | 1.947 | 6.258 | 0.520 | 1.639 | 0.575 | 1.560 | 0.629 | 2.253 | 0.543 | 1.674 |
| B8 800x1333 d256 L4 | FP16 | 1.755 | 4.882 | 0.618 | 1.719 | 0.593 | 1.599 | 0.589 | 2.311 | 0.565 | 1.539 |
| | BF16 | 1.721 | 7.468 | 0.602 | 1.701 | 0.564 | 1.581 | 0.560 | 3.175 | 0.544 | 2.043 |

centroids; (iv) our downstream primitive is argsort-dispatch of the stock kernel, whereas DANMP's is packing (a gather-plus-batch fusion of bilinears that share four-corner sub-targets). All four deviations make our test weaker at detecting the locality signal DANMP targets.

A partial port closing three of the four deviations (sampling-point synthesis from reference + uniform $\Delta P$, DBSCAN-style 9-pixel-radius variable-$k$, 20% query subsample) and retaining argsort-dispatch reproduces the H100 null cleanly (12/12 cells with $|r-1| \leq 2\%$) and shows no regression on 10/12 A100 cells under a one-sided regression test (latency ratio $\leq 1.02$); two A100 FP32 cells ($B=4$, $B=8$ decoder) exhibit a reproducible ~15–18% slowdown under uniformly-random $\Delta P$ synthesis. The nine A100 BF16/FP16 cells

Table 23: Full operator-level latency sweep on H100. $p_{50}$ in milliseconds, eager mode, external softmax.

| Config | Precision | Reference Fwd | Reference Bwd | CUDA (Zhu et al., 2021) Fwd | CUDA (Zhu et al., 2021) Bwd | CUDA *ours* Fwd | CUDA *ours* Bwd | Triton, point-parallel Fwd | Triton, point-parallel Bwd | Triton, query-block Fwd | Triton, query-block Bwd |
|---|---|---|---|---|---|---|---|---|---|---|---|
| | FP32 | 0.860 | 1.977 | 0.415 | 1.128 | 0.484 | 1.095 | 0.445 | 1.113 | 0.452 | 1.264 |
| B1 640 d256 L4 | FP16 | 0.855 | 1.869 | 0.435 | 1.231 | 0.457 | 1.267 | 0.426 | 1.126 | 0.437 | 1.149 |
| | BF16 | 0.850 | 1.858 | 0.434 | 1.228 | 0.458 | 1.160 | 0.428 | 1.126 | 0.434 | 1.145 |
| | FP32 | 1.086 | 2.383 | 0.408 | 1.070 | 0.476 | 1.139 | 0.472 | 1.151 | 0.451 | 1.173 |
| B2 1280 d256 L4 | FP16 | 1.021 | 2.112 | 0.444 | 1.176 | 0.468 | 1.206 | 0.463 | 1.243 | 0.442 | 1.179 |
| | BF16 | 1.018 | 2.099 | 0.443 | 1.170 | 0.467 | 1.198 | 0.465 | 1.249 | 0.441 | 1.180 |
| | FP32 | 1.245 | 3.289 | 0.414 | 1.073 | 0.479 | 1.238 | 0.475 | 1.154 | 0.454 | 1.173 |
| B2 1536x2048 d256 L4 | FP16 | 1.145 | 2.584 | 0.445 | 1.175 | 0.473 | 1.204 | 0.470 | 1.242 | 0.446 | 1.183 |
| | BF16 | 1.146 | 2.591 | 0.449 | 1.167 | 0.474 | 1.204 | 0.467 | 1.227 | 0.445 | 1.182 |
| | FP32 | 21.179 | 51.437 | 4.015 | 13.559 | 2.867 | 11.639 | 9.329 | 34.150 | 1.843 | 6.677 |
| B2 1536x2048 d256 L4 enc | FP16 | 16.484 | 49.477 | 3.966 | 12.983 | 1.653 | 10.169 | 10.045 | 56.283 | 1.143 | 4.309 |
| | BF16 | 16.488 | 48.983 | 3.968 | 12.986 | 1.657 | 10.188 | 9.982 | 50.662 | 1.151 | 4.334 |
| | FP32 | 0.780 | 1.626 | 0.411 | 1.056 | 0.481 | 1.126 | 0.448 | 1.254 | 0.456 | 1.285 |
| B2 800x1333 d256 L2 | FP16 | 0.761 | 1.623 | 0.444 | 1.145 | 0.468 | 1.179 | 0.435 | 1.261 | 0.444 | 1.290 |
| | BF16 | 0.760 | 1.618 | 0.448 | 1.148 | 0.471 | 1.177 | 0.434 | 1.260 | 0.441 | 1.286 |
| | FP32 | 0.988 | 2.199 | 0.412 | 1.143 | 0.476 | 1.118 | 0.447 | 1.130 | 0.451 | 1.156 |
| B2 800x1333 d256 L4 | FP16 | 0.974 | 2.207 | 0.446 | 1.149 | 0.461 | 1.173 | 0.435 | 1.140 | 0.440 | 1.157 |
| | BF16 | 0.971 | 2.207 | 0.445 | 1.144 | 0.467 | 1.172 | 0.434 | 1.139 | 0.442 | 1.154 |
| | FP32 | 7.202 | 18.344 | 1.325 | 4.308 | 0.731 | 3.637 | 3.176 | 11.588 | 0.647 | 2.387 |
| B2 800x1333 d256 L4 enc | FP16 | 5.638 | 18.427 | 1.276 | 4.074 | 0.502 | 3.283 | 3.362 | 63.864 | 0.477 | 1.571 |
| | BF16 | 5.644 | 18.360 | 1.275 | 4.075 | 0.503 | 3.303 | 3.341 | 25.184 | 0.477 | 1.570 |
| | FP32 | 1.397 | 4.114 | 0.411 | 1.060 | 0.482 | 1.120 | 0.449 | 1.140 | 0.456 | 1.164 |
| B2 800x1333 d256 L5 | FP16 | 1.333 | 3.187 | 0.445 | 1.262 | 0.469 | 1.188 | 0.435 | 1.149 | 0.444 | 1.175 |
| | BF16 | 1.326 | 3.199 | 0.446 | 1.159 | 0.469 | 1.187 | 0.434 | 1.148 | 0.445 | 1.176 |
| | FP32 | 1.147 | 2.632 | 0.439 | 1.113 | 0.508 | 1.179 | 0.476 | 1.196 | 0.482 | 1.218 |
| B2 800x1333 d512 L4 | FP16 | 1.087 | 2.385 | 0.444 | 1.158 | 0.468 | 1.191 | 0.436 | 1.147 | 0.443 | 1.176 |
| | BF16 | 1.087 | 2.372 | 0.445 | 1.153 | 0.470 | 1.188 | 0.438 | 1.147 | 0.446 | 1.178 |
| | FP32 | 1.091 | 2.537 | 0.409 | 1.070 | 0.478 | 1.236 | 0.448 | 1.144 | 0.454 | 1.170 |
| B4 800x1333 d256 L4 | FP16 | 1.055 | 2.150 | 0.447 | 1.170 | 0.465 | 1.306 | 0.440 | 1.163 | 0.445 | 1.185 |
| | BF16 | 1.053 | 2.142 | 0.446 | 1.170 | 0.472 | 1.206 | 0.441 | 1.160 | 0.445 | 1.187 |
| | FP32 | 14.021 | 35.468 | 2.607 | 8.466 | 1.411 | 7.078 | 6.308 | 22.952 | 1.231 | 4.600 |
| B4 800x1333 d256 L4 enc | FP16 | 10.974 | 35.854 | 2.552 | 8.044 | 0.943 | 6.460 | 6.737 | 126.566 | 0.782 | 3.057 |
| | BF16 | 10.986 | 35.886 | 2.553 | 8.045 | 0.947 | 6.468 | 6.702 | 50.998 | 0.786 | 3.062 |
| | FP32 | 1.315 | 4.107 | 0.413 | 1.163 | 0.476 | 1.129 | 0.475 | 1.450 | 0.454 | 1.167 |
| B8 800x1333 d256 L4 | FP16 | 1.272 | 3.100 | 0.470 | 1.272 | 0.470 | 1.198 | 0.460 | 1.735 | 0.441 | 1.183 |
| | BF16 | 1.274 | 3.101 | 0.470 | 1.269 | 0.473 | 1.195 | 0.462 | 1.730 | 0.442 | 1.177 |

Table 24: Numerical parity versus the FP32 reference (cast to float64). All backends pass a $10\times$ threshold test.

| Backend | FP32 max abs | FP32 mean rel | FP16 max abs | FP16 mean rel | BF16 max abs | BF16 mean rel |
|---|---|---|---|---|---|---|
| Reference | 0 | 0 | 0.367 | 0.225 | 0.573 | 1.49 |
| CUDA path | $1.6 \times 10^{-6}$ | $5.9 \times 10^{-6}$ | 0.366 | 0.221 | 0.561 | 1.46 |
| Triton path | $1.6 \times 10^{-6}$ | $5.8 \times 10^{-6}$ | 0.367 | 0.264 | 0.561 | 1.46 |

run $\sim 2\%$ faster under this reordering, but the magnitude is comparable to per-trial standard deviation and we do not interpret it as a meaningful speedup.

A follow-up control replaces the uniform-random offsets with offsets produced by an init-perturbed projection layer (Gaussian weights, $\sigma=0.01$), making them input-dependent. The A100 FP32 asymmetry vanishes: across five $B\in\{2,4,8\}$ BF16/FP32 cells, every cell falls within $\pm 0.2\%$ of linear (mean ratios 0.999–1.002, three-trial means). The A100 FP32 effect is therefore a property of the uniform-random $\Delta P$ synthesis distribution rather than of sampling-point clustering itself, and disappears under more realistic offsets.

# G  Supplementary Figures

This appendix collects the full set of generated benchmark visualizations. Figure 5 appears in the main body; Figures 9 and 10 are included here for completeness.

Table 25: Cumulative technique ablation at encoder scale ($B$=2, 1536×2048, $L$=4, BF16).

| Backend | Cumulative techniques | A100 | | H100 | |
|---|---|---|---|---|---|
| | | Fwd (ms) | Bwd (ms) | Fwd (ms) | Bwd (ms) |
| Reference impl. | (none) | 25.57 | 692.33 | 16.49 | 48.98 |
| CUDA baseline | rebuild only | 6.65 | 24.28 | 3.97 | 12.99 |
| CUDA + read-side | + vectorized loads, warp broadcast | 3.12 | 16.86 | 1.67 | 9.99 |
| CUDA + read + write | + relaxed atomics | 2.92 | 17.21 | 1.66 | 10.19 |
| Triton point-parallel | occupancy-maximizing tiling | 16.45 | 94.25 | 9.98 | 50.66 |
| Triton query-block | query-block tiling + relaxed ordering | 2.00 | 123.40 | 1.15 | 4.33 |

```
1   // Forward: each warp processes one (batch, query, head) tuple.
2   // Sampling coordinates are loaded once per warp by lane 0 and
3   // broadcast to the other 31 lanes via __shfl_sync.
4   __global__ void forward_kernel(
5       const scalar_t* __restrict__ p_value,  // feature maps
6       const scalar_t* __restrict__ p_loc,    // sampling coords (disjoint)
7       const scalar_t* __restrict__ p_attn,   // attention weights (disjoint)
8       const int64_t*  data_spatial_shapes,
9       const int64_t*  data_level_start_index,
10      scalar_t*       p_output) {
11
12    // Cache attention weights for this (query, head) in shared memory
13    // once; every lane reads its LK values locally thereafter.
14    __shared__ scalar_t p_mask_shm[L * K];
15    if (threadIdx.x < L * K)
16      p_mask_shm[threadIdx.x] = p_attn_ptr[threadIdx.x];
17    __syncwarp();
18
19    const uint32_t lane_mask = 0xFFFFFFFF;
20    for (int li = 0; li < L; ++li) {
21      const int spatial_h = data_spatial_shapes[li * 2];
22      const int spatial_w = data_spatial_shapes[li * 2 + 1];
23      for (int ki = 0; ki < K; ++ki) {
24
25        // Lane 0 issues the single global load for this sample point;
26        // all 32 lanes of the warp then read the same (loc_w, loc_h)
27        // via a register-to-register shuffle.
28        opmath_t loc_w, loc_h;
29        if (threadIdx.x == 0) {
30          loc_w = (opmath_t)p_loc_ptr[li * K * 2 + ki * 2    ];
31          loc_h = (opmath_t)p_loc_ptr[li * K * 2 + ki * 2 + 1];
32        }
33        loc_w = __shfl_sync(lane_mask, loc_w, 0);
34        loc_h = __shfl_sync(lane_mask, loc_h, 0);
35
36        const opmath_t attn = p_mask_shm[li * K + ki];
37        const opmath_t w_im = loc_w * spatial_w - 0.5;
38        const opmath_t h_im = loc_h * spatial_h - 0.5;
39
40        // Each lane is responsible for a disjoint channel slice.
41        // Wide (ulonglong4) loads on the feature row below.
42        accumulate_bilinear_sample(p_value, spatial_h, spatial_w,
43                                   w_im, h_im, attn, p_output);
44      }
45    }
46  }
```

Figure 7: Warp-level sharing of sampling coordinates in the CUDA forward kernel. Lane 0 issues a single 16 byte load for each sampling-coordinate pair per sample point and broadcasts it to the other 31 lanes of the warp via a warp shuffle, amortizing one coordinate fetch across the whole warp. Disjoint sampling-location and attention-weight pointers eliminate the concatenated sampling-parameter buffer of the reference implementation (Section 2.3).

```
1   # Backward: each program processes one (batch, query, head) tile
2   # of channels. Gradient writes to grad_value use sem="relaxed"
3   # atomics to skip the L1 coherence sectors that would otherwise
4   # serialize contending warps at the same cache line.
5   @triton.jit
6   def backward_kernel(
7       value_ptr, loc_ptr, attn_ptr,
8       grad_out_ptr, grad_value_ptr, grad_loc_ptr, grad_attn_ptr,
9       # ... strides and shapes elided ...
10  ):
11      b  = tl.program_id(0)
12      q  = tl.program_id(1)
13      g  = tl.program_id(2)
14
15      # For each of the L * K sample points in this (batch, query, head)
16      # tile, compute the bilinear neighbour offsets and gradient share.
17      for lk in range(L_TIMES_K):
18          loc_w = tl.load(loc_ptr + q * stride_locq + lk * 2     )
19          loc_h = tl.load(loc_ptr + q * stride_locq + lk * 2 + 1)
20          attn  = tl.load(attn_ptr + q * stride_attnq + lk)
21
22          wx0, wy0, wx1, wy1, p00, p01, p10, p11, m00, m01, m10, m11 = \
23              bilinear_neighbours(loc_w, loc_h, spatial_shape(lk))
24
25          weighted_grad = grad_out * attn
26
27          # Four relaxed atomic scatter-accumulations per neighbour.
28          gv = grad_value_ptr
29          tl.atomic_add(gv + p00 * stride_gvs + d * stride_gvd,
30                        (weighted_grad * wx0 * wy0).to(gv.dtype.element_ty),
31                        mask=m00, sem="relaxed")
32          tl.atomic_add(gv + p01 * stride_gvs + d * stride_gvd,
33                        (weighted_grad * wx1 * wy0).to(gv.dtype.element_ty),
34                        mask=m01, sem="relaxed")
35          tl.atomic_add(gv + p10 * stride_gvs + d * stride_gvd,
36                        (weighted_grad * wx0 * wy1).to(gv.dtype.element_ty),
37                        mask=m10, sem="relaxed")
38          tl.atomic_add(gv + p11 * stride_gvs + d * stride_gvd,
39                        (weighted_grad * wx1 * wy1).to(gv.dtype.element_ty),
40                        mask=m11, sem="relaxed")
41
42          # grad_loc and grad_attn write to disjoint output tensors;
43          # no concatenated gradient buffer is materialized.
44          tl.store(grad_loc_ptr  + q * stride_glq + lk * 2,     grad_w)
45          tl.store(grad_loc_ptr  + q * stride_glq + lk * 2 + 1, grad_h)
46          tl.store(grad_attn_ptr + q * stride_gaq + lk,         grad_a)
```

Figure 8: Relaxed-ordering scattered atomic accumulation in the Triton backward kernel. Relaxed memory-ordering semantics on the atomic-addition intrinsic instruct the Triton compiler to elide the L1 coherence-sector traffic that the default acquire-release semantics would require around each transaction, which is safe for gradient accumulation because only the final value is observed. On SM 8.0 (A100) the relaxed path is necessary but not sufficient: the software compare-and-swap loop for BF16 still dominates at encoder scale (Section 3.3). On SM 9.0 (H100) the relaxed path combined with hardware BF16 atomics recovers full memory-pipeline throughput.

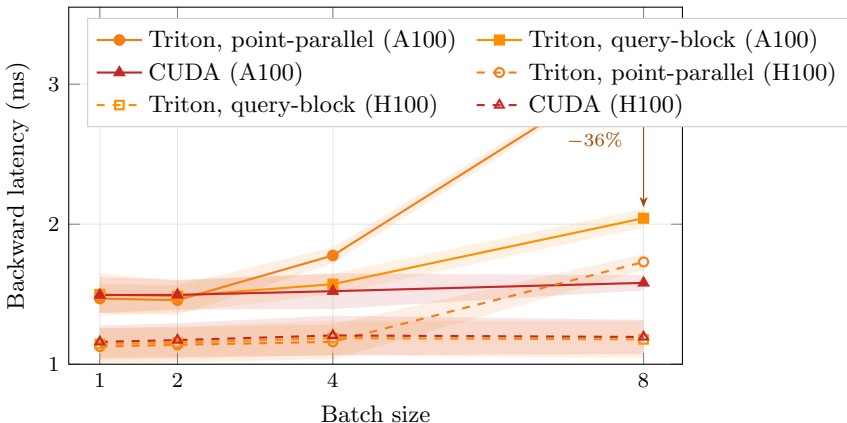

Figure 9: Backward-pass latency as a function of batch size at BF16, decoder configuration. Solid lines denote A100; dashed lines denote H100. The Triton point-parallel tiling (orange circles) diverges sharply from the query-block tiling (orange squares) at $B{\geq}4$ on A100, reflecting the atomic-contention scaling analyzed in Section 3.3.

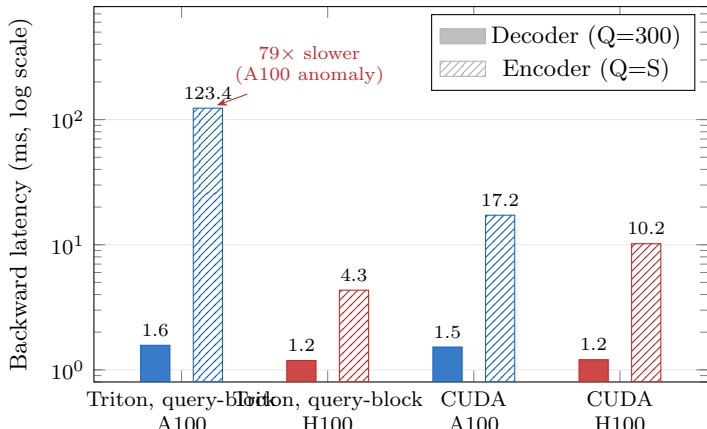

Figure 10: Decoder-scale versus encoder-scale backward latency (BF16, log scale) for both programming models on both GPUs. At encoder scale on A100, the Triton query-block backward takes 123.4 ms ($79\times$ its decoder-scale value), while the CUDA backward takes 17.2 ms. On H100 the Triton kernel recovers to 4.3 ms, faster than the CUDA path (10.2 ms), illustrating the architectural crossover analyzed in Sections 3.3 and 4.1.

