# OpenReview forum: "Efficient Multi-Scale Deformable Attention on GPUs"
_TMLR — Under review for TMLR_

### Review · Reviewer_THKC · 2026-05-26

**Summary Of Contributions:**

This paper addresses the core bottleneck of the DETR series models, Multiscale Deformable Attention (MSDA), and conducts a systematic hardware-level diagnostic study on two generations of NVIDIA A100/H100 GPUs. It reveals three fundamental failure modes of traditional GPU optimization heuristics on the scattering access operator, accurately pinpoints the hardware-compiler co-factor of the backpropagation performance gap, and provides an optimal cross-architecture implementation scheme. It achieves a 2.4-14x forward speedup and up to 88% reduction in memory usage.

**Audience:**

Yes

**Audience Explanation:**

This paper adopts a research method based on hardware counters, and systematically disproved two widely accepted GPU optimization dogmas through rigorous controlled variable experiments and proof by contradiction.

**Claims And Evidence:**

Yes

**Claims Explanation:**

The paper constructs a comprehensive experimental evaluation system, covering 7 query scheduling orders, 2 programming models (CUDA/Triton), 3 precision levels (FP32/FP16/BF16), and 12 different operator configurations (including encoder/decoder scales, different feature layers, and different hidden dimensions). It also provides detailed hardware counter data, roofline analysis, technical ablation, and numerical precision verification. All experimental results are accompanied by standard deviations and confidence intervals, demonstrating clear statistical significance.

**Requested Changes:**

1. The scope of application of the core conclusions is unclear, posing a risk of overgeneralization.
The paper's conclusion that "query scheduling order reordering is ineffective" is only validated under the standard configuration of K=4, L=4, and does not test cases with larger K values ​​(e.g., K=8, K=16) or more feature layers.

2. Key baseline comparisons are severely lacking, making it impossible to accurately assess the relative magnitude of the contributions.
Comparisons with the MSDA implementation in the xFormers library are not included. xFormers is the most widely used high-performance operator library in the PyTorch ecosystem, and its MSDA implementation has been adopted by numerous industrial projects. The lack of this comparison makes the performance improvement data in this paper lack practical reference value.

3. The paper lacks end-to-end performance demonstration.
The paper only provides operator-level latency and memory data, failing to demonstrate the end-to-end training and inference acceleration of the complete model on standard tasks such as object detection and segmentation based on existing DETR-based methods. Especially regarding the 14x speedup at the encoder scale, the paper does not explain how much this acceleration reduces the overall training time in full Deformable DETR training.

4. The discussion of the FlashAttention series of works is inaccurate. The paper claims that FlashAttention's tiling method is not applicable to MSDA, but it fails to compare the performance of the optimized MSDA on downstream tasks with that of full attention based on FlashAttention.

---

> ### Author Response · Authors · 2026-07-06
> **Requested changes addressed**
>
> Thank you for the careful reading and for recognizing the diagnostic contribution. All four requested changes are addressed in the revision; section, figure, and table numbers refer to it.
>
> **1. Scope of the dispatch-order null (larger $K$, more levels).** The sweep now runs the seven orders at $K \in \{4,8,16\}$ and $L \in \{2,3,5\}$ on both GPUs (Tables 12-14); the abstract and Sections 1, 3.1, and 7 state the widened scope. The null holds throughout: at $K=8$ every batch $\geq 2$ cell is within 1.5% (encoder 0.2-0.4%); at $K=16$ the worst case is 1.58% (H100, FP32, batch 2); in the $L$-sweep all batch $\geq 2$ decoder cells are within 1.33%, apart from one flagged jitter outlier (Table 14). The mechanism predicts this $K$- and $L$-independence: the sampled addresses are data-dependent and effectively random under any static permutation, so no reordering manufactures reuse the query-block tiling does not already provide. One honest exception is disclosed rather than averaged away: at $L=5$ with full encoder resolution ($1536 \times 2048$, $N \approx 262$k), spatial orderings run 7-10% faster than linear (Hilbert: 9.8% H100, 7.4% A100), a genuine large-$N$ cache-locality effect at a different operating point (Table 14 caption) that does not reopen the decoder-scale invariance claim.
>
> **2. Baseline comparison.** One point may be worth clarifying first: xFormers does not currently provide a multi-scale deformable attention operator; its high-performance kernels target dense and block-sparse self-attention, so there is no MSDA operator to benchmark directly. (The "xMSDA" work you may be referring to is an Ascend-NPU implementation, Huang et al. 2025, which the paper already cites.) The implementations practitioners actually run are those shipped by the Transformers library (a native-PyTorch bilinear-sampling path and a compiled CUDA kernel) and the mmcv port of the Zhu et al. (2020) kernel the paper already benchmarks. The revision benchmarks the shipped Transformers operators directly (Table 17): at the encoder operating points our Triton kernel is 3-5$\times$ faster than the compiled CUDA kernel (up to 5.1$\times$ at Cityscapes) and 11-21$\times$ faster than the native-PyTorch path, and is more accurate (BF16 matches an FP32 ground truth everywhere, while the shipped CUDA kernel reaches a maximum relative error of 266 on captured inputs, Table 9). Section 6 now states what xFormers contains.
>
> **3. End-to-end performance and training-time share.** The revision swaps our kernels into stock Deformable-DETR and Mask2Former (pretrained weights, parity-checked against FP32) and measures whole-model step time and MSDA's wall-clock share (Table 8): MSDA is 63-65% of forward and 78-96% of backward in Deformable-DETR (32-34% / 61-95% for Mask2Former, whose backbone dilutes the share). Over a COCO epoch (Table 18), the recommended kernel cuts Deformable-DETR from 1.22 to 0.98 h/epoch on H100 (-20%) and Mask2Former from 1.47 to 1.28 h (-13%); on A100 the reductions are -9% and -5%. The $14\times$ operator-level encoder speedup does not translate 1:1: the backbone and dense heads bound the reduction (Amdahl). The central finding reproduces at model scale: relative to the Zhu et al. kernel the Triton backward is $\sim 10\times$ slower on A100 but 1.2-1.3$\times$ faster on H100, so the winning implementation flips with GPU generation end to end. At the operating points (Table 15) the operator-level advantage is larger, e.g. $1.89\times$ forward / $3.02\times$ backward over Zhu et al. on H100 at Cityscapes encoder scale, with the FP32-accumulator variant repairing the A100 backward ($1.29\times$ vs. $0.11\times$). Memory: 448 MB at encoder-scale backward vs. 3608 MB native-PyTorch (-88%) and 1056 MB optimized CUDA (Table 7). Whole-model torch.compile buys only $\sim 1.1\times$ forward and cannot touch the operator; a captured-replay study on real images (Table 9) confirms 2.9-6.0$\times$ over Zhu et al. and 2.6-4.9$\times$ over the Transformers CUDA kernel at FP32 parity.
>
> **4. FlashAttention discussion.** Section 6 now tightens the claim and makes it quantitative: FlashAttention's contiguous-tile structure does not transfer to MSDA's data-dependent scattered gather, not that no idea from that line applies. The requested comparison (FlashAttention full attention vs. optimized MSDA) is in effect DETR vs. Deformable-DETR, the dense $O(N^2)$ model MSDA was introduced to replace at encoder token counts. The revision measures it (Table 17): at COCO encoder scale ($N \approx 22$k) full self-attention via FlashAttention is 12-15$\times$ slower than our Triton MSDA, and at Cityscapes ($N \approx 44$k) 29-38$\times$ slower with higher memory; even without the memory wall the $O(N^2)$ compute remains.

---

### Review · Reviewer_P4TP · 2026-06-20

**Summary Of Contributions:**

This paper presents a diagnostic study of GPU kernel optimization for multi-scale deformable attention on A100 and H100 GPUs. The main claims are that query-dispatch reordering does not improve forward latency in the tested setting, high occupancy is not a reliable proxy for throughput, and BF16 backward performance is strongly affected by hardware/compiler support for atomic additions.

**Audience:**

Yes

**Audience Explanation:**

Yes. The findings would be interesting to researchers working on  DETR-style deformable attention, and GPU kernel optimization.

**Broader Impact Concerns:**

I do not see major broader-impact concerns specific to this work. The paper is mainly about improving the efficiency of an existing operator, which may reduce compute and memory requirements. A short statement on compute/energy implications would be sufficient.

**Claims And Evidence:**

Yes

**Claims Explanation:**

Partially, but not sufficiently in the current form. The paper contains some strong operator-level evidence, especially for the BF16 backward bottleneck: the timing results are connected to hardware atomic support, compiler lowering, SASS-level inspection, and hardware counters. The occupancy result is also reasonably convincing.

However, the paper’s broader conclusions are harder to accept as stated. The experiments are mostly limited to two NVIDIA GPU generations and one CUDA/PyTorch/Triton stack, even though the main claims depend strongly on hardware and compiler behavior. The connection to real model behavior is also underdeveloped: the paper reports operator-level benchmarks, but not end-to-end training throughput, memory, or convergence behavior. Finally, the paper is not clear enough about the scope of each conclusion. The reordering experiment, in particular, is difficult to evaluate because the tested orderings are not defined clearly enough in the main text (though some details given in the supplemental),

Overall, I find the evidence promising but not yet clear or broad enough for acceptance.

**Requested Changes:**

It would be great if the authors can summarize their findings/conclusions in bullets in the introduction, and state clearly the sections for the experiments to prove the claims. The current form is a bit messy and hard for someone not familiar with the field to quickly capture the main claims of the paper.

It would also be good to improve the presentation of the reordering experiment, including precise definitions or pseudocode for the tested orderings and conceptual diagrams for MSDA sampling, query ordering, query-block vs. point-parallel tiling, and backward atomic accumulation.

---

> ### Author Response · Authors · 2026-07-06
> **Response and revision**
>
> Thank you for the careful reading and for judging the operator-level evidence (the BF16 backward bottleneck and the occupancy result) strong. This response addresses the three substantive concerns, then the two Requested Changes and broader impact. Section, figure, and table numbers refer to the revised manuscript.
>
> **Generality: "two NVIDIA generations, one stack."** The two generations are not a coverage sample but the controlled experiment that isolates the backward bottleneck's cause: the finding is precisely that the result *flips* between A100 (SM 8.0, no native BF16 atomic add, emulated as a compare-and-swap retry loop) and H100 (SM 9.0, native L2 reduction). Holding the kernel fixed and varying only the architecture licenses attributing the gap to hardware atomic support rather than the kernel; the companion compiler-gated finding (the lowering flips *within* Hopper by compiler path) makes the same isolation argument on the toolchain axis. The revision states this design and adds the direct counter measurement that closes the causal chain (Table 3: the A100 BF16 path issues 34.8M compare-and-swap sectors and no hardware reductions; every other configuration is the reverse). Section 7 now frames the claim at the portable mechanism (architectures lacking a native low-precision atomic serialize under contention), with other vendors as future work.
>
> **Real-model behavior.** The revision adds a full end-to-end study: kernels swapped into stock Deformable-DETR and Mask2Former with pretrained weights, measuring whole-model step time and MSDA's wall-clock share (Table 8; MSDA is 63-65% of forward and 78-96% of backward for Deformable-DETR). Per-epoch training wall-clock (Table 18): on H100 the recommended Triton kernel cuts a COCO epoch from 1.22 to 0.98 h for Deformable-DETR (-20%) and 1.47 to 1.28 h for Mask2Former (-13%); on A100 the recommended CUDA kernel cuts 1.90 to 1.73 h (-9%) and 2.75 to 2.62 h (-5%), versus the Zhu et al. kernel. Encoder-scale memory (Table 7): 447.9 vs. 3607.8 MB (-88%). On convergence, the study does not retrain and now makes the argument explicit: the kernels match the FP32 reference to tolerance on the exact inputs a trained model produces (captured real-image replay, Table 9; synthetic parity, Table 24), so released-checkpoint metrics are unchanged by construction; indeed our kernels are *more* accurate than the shipped production kernels. The one residual training-time concern, the native-BF16 backward's relaxed-ordering nondeterminism, is now quantified (Table 10): $4\times10^{-5}$ to $3\times10^{-4}$ relative $L_2$, about four orders of magnitude below minibatch gradient variation ($\approx 1.4$); the FP32-accumulator variant is deterministic to $\sim 10^{-7}$.
>
> **Scope clarity.** The revision tightens scope at each claim: the dispatch-order null is scoped to the decoder operating point across $K \in \{4,8,16\}$ and $L \in \{2,3,5\}$ (from the single $K=4$/$L=4$), every batch $\geq 2$ cell within $\pm 2\%$ on both GPUs apart from one flagged jitter outlier (Table 14; Section 3.1, Tables 12-14). The one honest exception is disclosed: at $L=5$ with full encoder resolution ($1536 \times 2048$, $N \approx 262$k), cache-friendly orderings run 7-10% faster, a distinct large-$N$ effect outside the invariance claim's operating points. Speedup claims are split by scale (decoder vs. encoder) and backward claims tagged with their SM-version dependence.
>
> **Requested Change 1: bulleted findings with section pointers.** Done. Three bulleted contributions, each stating scope and pointing to its section and table: the dispatch-order null (Section 3.1, Table 11); the occupancy-throughput inversion (85% occupancy at 5.1% of peak bandwidth vs. a 17%-occupancy tiling $7.4\times$ faster, Section 3.2, Table 1); the hardware-gated BF16 backward with the FP32-accumulator mitigation (Section 3.3, Table 8). A closing sentence adds the open-source library and its end-to-end and training results (Section 5, Tables 8 and 18).
>
> **Requested Change 2: reordering presentation and diagrams.** Done. The seven orderings are defined in the main text by explicit sort keys (a Section 3.1 display equation: identity, scanline rank, Morton Z-order, Hilbert index, reference-point centroid, uniform-random, and the K-means clustering-and-packing analogue of DANMP); the four-axis DANMP port stays in Appendix F. Four schematics are added: MSDA's scattered multi-scale sampling (Figure 1), query dispatch ordering (Figure 2), query-block vs. point-parallel tiling with occupancy/bandwidth annotations (Figure 3), and backward atomic accumulation (CAS loop on SM 8.0 vs. native reduction on SM 9.0, Figure 4).
>
> **Broader impact.** Added (Section 7): the work improves an existing operator's efficiency, introduces no new capability, and leaves outputs numerically unchanged; the concrete benefit is the per-epoch reductions above at no accuracy cost.

---

### Review · Reviewer_qfom · 2026-06-22

**Summary Of Contributions:**

This is a diagnostic GPU-kernel study (the authors are explicit that "the kernel techniques themselves
  are standard," p.2) of the multi-scale deformable attention (MSDA) operator that underpins DETR-family
  detectors and segmenters (Deformable DETR, DINO, Mask2Former). Working on A100 (SM 8.0) and H100 (SM
  9.0), the paper characterizes why MSDA — whose data-dependent, scattered bilinear sampling defeats
  FlashAttention-style tiling — resists the standard memory-bound-kernel playbook, and contributes:

  1. Two documented "failure modes" of conventional optimization heuristics.
    - Locality null: seven query-dispatch orders (linear, Morton, random, scanline, Hilbert, centroid,
  and a K-means clustering-and-packing analogue of DANMP) reportedly produce near-identical forward
  latency, because L2 locality is set by the query-block tiling, not by the global dispatch order; a
  uniform-random permutation (a lower bound on any structured ordering) matches linear.
    - Occupancy–bandwidth inversion: an 85%-occupancy point-parallel tiling delivers only 5.1% of A100
  peak bandwidth, while a 17%-occupancy query-block tiling delivers 36.3% and runs 7.4× faster (Table 1),
  with warp-stall attribution (67.2% long-scoreboard) tying the inversion to DRAM transaction size, not
  warp availability.
  2. A hardware-and-compiler-gated root cause for the backward-pass bottleneck, with a mitigation. The
  scattered-gradient atomic accumulation collapses to 2.4–2.5% of A100 peak bandwidth at BF16 vs 21.3% on
  H100 on bit-identical code (Table 2). The mechanism is evidenced at the SASS level: H100/Triton lowers
  to a single REDG.E.ADD.BF16x8.RN.STRONG.GPU reduction while A100 (and nvcc's CUDA-12.8 half-precision
  overload, even on H100) lowers to an ATOM.E.CAS retry loop. An FP32-accumulator variant closes the A100
  gap to within 6% of the CUDA path (Table 3: 18.2 vs 17.2 ms), reframing the backward choice as an
  accumulator-precision decision gated by target hardware. A 72-cell PyTorch-native scatter-add sweep is
  used as a falsification control showing the gap does not reproduce on a different compiler path (Tables
  8–9).
  3. An open-source CUDA + Triton library reporting up to ~14× forward speedup (encoder scale) and up to
  88% peak-VRAM reduction over the Zhu et al. (2021) reference, at claimed numerical parity (Table 12).

  Key strengths
  - Headline numbers are internally reproducible from the tables to within rounding (independently
  recomputed: 0.612/0.083 = 7.37× ≈ 7.4×; 16.616/1.367 = 12.15× ≈ 12.18×; 10.986/0.786 = 13.98× ≈ 14.00×;
  18.2/17.2 = 5.8% < 6%).
  - The backward-pass root cause is mechanistically grounded (matched-code cross-GPU design + SASS/PTX
  fingerprints + a purpose-built native-BF16 control backend at 1.706 ms that splits the gap into
  precision-rescue vs codegen components), not a black-box latency argument.
  - The scatter-add falsification experiment is a genuine attempt to break the authors' own claim —
  unusually rigorous.
  - Sound, disclosed measurement protocol (p50 over n=100 after 10 warmups with device-sync barriers;
  absolute latencies alongside ratios to avoid the A100→H100 baseline confound; kernel-replay counter
  caveat correctly scoped to the SASS fingerprint, not to latencies).
  - Candid scoping: kernel techniques labeled "standard," the DANMP/CAP port's four-axis deviation
  disclosed (Appendix F), and the no-training-curve / inference-only-TensorRT / two-NVIDIA-GPU
  limitations stated (p.12).

  Key weaknesses
  - Several headline claims currently overstate or are mis-scoped relative to the paper's own tables (the
  categorical "within ±2%", the "2.4–14×" scale conflation, the un-tabulated "88%", the overloaded term
  "numerical parity").
  - Statistical support for the locality null is uneven: a single aggregate minimum-detectable-effect
  (MDE) does not cover the high-dispersion cells, and there is no per-cell equivalence test/CI.
  - Sole speedup baseline is the dated 2021 reference kernel; the self-named "closest peer" (DCNv4) is
  never benchmarked.
  - No code at review time, and no training-time / training-curve / task-accuracy numbers despite a
  training-motivated framing; the nondeterministic relaxed-ordering BF16 backward is not validated for
  training convergence.

**Audience:**

Yes

**Audience Explanation:**

There is a well-defined and non-trivial audience: (a) practitioners training/deploying
  DETR-family detectors and segmenters (Deformable DETR, DINO, Mask2Former), for whom MSDA is an           established bottleneck (49% of GFLOPs in the Deformable DETR encoder); (b) GPU-kernel and compiler
  engineers; and (c) the systems-for-ML community.

  The findings are concrete and actionable:
- The negative result that query-dispatch reordering (Morton/Hilbert/K-means/DANMP-style) does not help
  on commodity GPUs saves others wasted effort and contradicts a CPU-side positive result from prior
  work
— exactly the kind of rigorous negative/diagnostic result TMLR explicitly welcomes.
- The occupancy ≠ throughput inversion is a counterintuitive, transferable lesson for kernel tuning of
  scattered-access operators.
- The SM 8.0 vs SM 9.0 BF16-atomic boundary with a clear FP32-accumulator recommendation is a genuinely
  useful, immediately usable systems insight, sharpened by SASS-level evidence and a cross-compiler-path   control.
- An open CUDA+Triton library (promised) increases practical relevance.

**Broader Impact Concerns:**

not relevant

**Claims And Evidence:**

Yes

**Claims Explanation:**

The three core diagnostic mechanisms are well-supported; several headline/abstract claims are currently over-stated or mis-scoped relative to the paper's own evidence and should be revised before they can be considered "supported."

Well-supported claims :
  - Occupancy–bandwidth inversion — directly evidenced by Table 1 and corroborated by warp-stall
  attribution.
  - Hardware-gated A100/H100 backward collapse on matched code — Table 2 + bit-identical kernel + SASS
  fingerprints + the native-BF16 control backend. This is the paper's strongest result.
  - FP32-accumulator closes the A100 gap — Table 3 (within 6%); arithmetic verified.
  - Compiler-path specificity — the scatter-add control (Tables 8–9) genuinely shows the 7× gap does not
  reproduce on the nvcc path; the atom.*↔red.* sector flip is concrete.


Claims that currently overstate or are mis-scoped:

1. The categorical "within ±2%" locality null is contradicted by the paper's own Table 6. The abstract
  states all seven orders "produce within-±2% forward latency," and §3.1 restates it (scoped to BF16) as
  "every reordering's p50 latency is within ±2%." But Table 6 shows the two locality-faithful orders
  exceed the band: danmp_partial r = 1.2058 (+20.6%) and kmeans_cap r = 1.1322 (+13.2%) at A100 B4/FP32 —
  and these are low-dispersion cells (σ/μ ≈ 1.2%), so they are real, well-resolved violations, not
  noise. Even within the BF16 scope §3.1 invokes, kmeans_cap on A100 is +3.9% (B4) and +4.5% (B8). The
  categorical wording therefore outruns the data for exactly the orders meant to reproduce the prior
  locality method. (Note: my initial draft also flagged the high-σ/μ B2/FP16 kmeans_cap/danmp_partial
  cells as "contradicting" the band — on verification that was a misreading; those cells are within the
  band with merely wide error bars. The genuine contradiction is the tight-dispersion B4/B8 cells above.)
  The mechanism argument may still be correct, and the authors do hedge in footnote 1 and Appendix F —
  but the abstract/§3.1 claim must be hedged to match.
  2. Statistical licensing of the null is incomplete. The MDE (~0.32%/0.36%, footnote 1) is correctly
  computed but only from the median (well-behaved) baseline σ; it does not cover cells where σ/μ reaches
  50–75% (e.g. A100 B2/FP16 kmeans_cap σ/μ = 64.84%; H100 B2/FP16 = 75.41%; H100 B8/FP16 random =
  49.84%). For those cells a 2% effect cannot be excluded at 95% (CI half-width ≈ 13%). There is no
  per-cell equivalence test (TOST), CI on r, or bootstrap, and Table 6 reports no p99 for the dispatch
  orders, so those high-variance cells are flagged (†) but never diagnosed as genuine reordering
  instability vs measurement contamination. Use of the robust median p50 softens this but does not close
  it.
  3. "2.4–14× forward speedup" silently conflates two scales. The 14× is encoder-scale H100 (14.00×,
  p.9); decoder-scale forward is 2.39–2.68× (Table 4). The intro is careful ("median 2.4–2.7× at decoder,
  12–14× at encoder"), but the abstract collapses these into one continuous range; moreover the 2.4×
  lower bound matches no decoder cell (the true decoder minimum is H100 CUDA 2.26×, below 2.4×). A
  fixable wording issue.
  4. "Up to 88% peak VRAM reduction at numerical parity." The 88% is derivable (encoder-scale prose:
  3607.8 → 447.9 MB = 87.6%) but appears in no table; the only memory table (Table 5) is decoder-scale
  and shows fwd+bwd peak unchanged at 151.7 MB. On verification, the "tension with the parity variant
  costing +29% VRAM" charge is refuted: the native-BF16 path that achieves 88% does pass the paper's
  tabulated 10× parity test (Table 12), so "at numerical parity" is defensible. The remaining fair issues
  are (a) the 88% should be tabulated, and (b) "numerical parity" is used ambiguously — p.7 applies the
  same phrase to the FP32-accumulator variant (which costs +29% more VRAM), inviting misreading.
  5. The "hardware-AND-compiler-gated" decomposition is partly asserted. Toolkit-version dependence is
  openly disclosed (CUDA 12.8). But no experiment isolates the relaxed-ordering qualifier from nvcc's
  no-return-overload promotion to prove which is the decisive H100 lever; indeed the text's own wording
  ("return is dead, which the PTX assembler promotes…") suggests the no-return overload, and Table 13
  shows CUDA relaxed atomics do not reproduce the reduction lowering. The A100 "serialization dominates"
  attribution is well-corroborated by the Table 8 collision-count sweep and FP32-indifference variance
  signature, but no direct retry-count counter is shown.
  6. Headline–baseline gap. Every speedup is against Zhu et al. (2021) only. DCNv4 — called "the closest
  systems-level peer" (p.10) — is never benchmarked; orthogonality/combinability is asserted (citing
  DCNv4's own Table 10), not measured. This dents the "Efficient" framing more than the diagnostic claims
  (the paper does also report a matched-code internal CUDA-vs-Triton comparison), but a reader cannot
  tell whether DCNv4 is faster, slower, or combinable.
  7. Motivation vs evidence mismatch (training). The work is motivated by the encoder's
  compute/efficiency bottleneck (49% GFLOPs / 11% AP), yet all evidence is inference-only operator
  micro-benchmarks; there are no training-time, wall-clock-training, or training-curve numbers, and the
  COCO/Cityscapes/ADE20K "within rounding" task-parity claim has no mAP/mIoU values and no end-to-end
  retraining. Operator-level numerical parity does exist (Table 12, fwd+bwd), so the backward path is
  numerically validated — but the trained-accuracy effect of the nondeterministic relaxed-ordering BF16
  backward is untested.
  8. Minor numerical inconsistency. The most-repeated headline number, A100 backward BF16 bandwidth, is
  "2.4%" in the abstract/intro/conclusion/Fig 1/Table 2 caption but "2.5%" in the Table 2 body (38.5/1560
  = 2.47% → 2.5%, so the body is correct).

**Requested Changes:**

Reconcile the categorical "within ±2%" locality null with your own Table 6.
The abstract ("seven query orders … produce within-±2% forward latency") and §3.1 ("every reordering's
  p50 latency is within ±2%") are contradicted by Table 6: danmp_partial r = 1.2058 (+20.6%) and
  kmeans_cap r = 1.1322 (+13.2%) at A100 B4/FP32 — both in tight-dispersion cells (σ/μ ≈ 1.2%), so these
  are real violations, not noise; even within the BF16 scope §3.1 invokes, kmeans_cap on A100 is +3.9%
  (B4) and +4.5% (B8). Reword to state precisely which orders × precisions × configs the ±2% holds for (it
  cleanly holds for the geometric orders — Morton/scanline/Hilbert/centroid — and for H100), and move the
  CAP/DANMP exceptions into the main-text claim rather than only footnote 1 / Appendix F. No new
  experiments required.

Provide statistical licensing for the null in the cells that matter.
  The MDE (~0.32%/0.36%, footnote 1) is computed only from the median, well-behaved baseline σ and does
  not cover cells with σ/μ = 50–75% (A100 B2/FP16 kmeans_cap 64.84%; H100 B2/FP16 75.41%; H100 B8/FP16
  random 49.84%), where a 2% effect cannot be excluded at 95% (CI half-width ≈ 13%). There is no per-cell
  CI/equivalence test (TOST) on r and no p99 for the dispatch orders.
  Add a per-cell CI on r (derivable from the σ you already report) or a TOST, or explicitly
  restrict the null claim to the low-dispersion regime it supports and flag the high-σ cells as
  undetermined; add p99 for the dispatch orders, or state why p50 alone suffices.

De-conflate the "2.4–14×" forward-speedup range in the abstract.
  14× is encoder-scale H100 (14.00×); decoder-scale forward is 2.39–2.68× (Table 4), and the 2.4× lower
  bound matches no decoder cell (true decoder min = H100 CUDA 2.26×).
  Carry the intro's scale qualifier ("median 2.4–2.7× at decoder, 12–14× at encoder") into
  the abstract.
Tabulate the "88% VRAM reduction" and disambiguate "numerical parity."
  The 88% appears in no table (only encoder-scale prose, 3607.8 → 447.9 MB); the only memory table (Table
  5) is decoder-scale and shows fwd+bwd peak unchanged at 151.7 MB. Separately, "numerical parity" is
  used for both the native-BF16 path (achieves 88%, passes Table 12's 10× test) and the FP32-accumulator
  path (p.7, which costs +29% VRAM) — inviting misreading.
  Add an encoder-scale memory table; state explicitly that the 88% is the native-BF16 path
  and that it satisfies the Table 12 parity criterion; reserve distinct wording for the FP32-accumulator
  property.

Fix the 2.4% vs 2.5% inconsistency (trivial but factual).
  The most-repeated headline number (A100 backward BF16 bandwidth) is "2.4%" in the
  abstract/intro/conclusion/Fig 1/Table 2 caption but "2.5%" in the Table 2 body (38.5/1560 = 2.47% →
  2.5%). Make it consistent.

---

> ### Author Response · Authors · 2026-07-06
> **Changes and corrections implemented**
>
> Thank you for an exceptionally careful and technical reading, including the independent recomputation of the headline ratios, and for judging the three core diagnostic mechanisms well-supported. All five Requested Changes are implemented; none required new experiments, except that investigating Change 1 uncovered a measurement artifact behind the contradicting cells, corrected here with a re-run. Table numbers refer to the revised manuscript; where the review cites a submitted-version table, the new number is given in parentheses.
>
> **1. Reconcile the "within $\pm 2\%$" null with Table 6 (now Table 11).** The submitted table did contradict the categorical claim, and the cause is a run-to-run measurement artifact, not a dispatch-order effect: the flagged A100 FP32 cells (partial-DANMP +20.6%, CAP +13.2%, batch 4) show no such deviation in the new $K=8$/$K=16$ sweeps (Tables 12-13), and a genuine locality penalty cannot be $K$- and precision-specific since the sampled addresses are identical across dtypes; re-running the original $K=4$ sweep with the corrected harness makes them vanish (batch-4 CAP $r=1.001$, partial-DANMP $r=1.0005$, both $\approx 1.000$ at batch 8). Table 11 is regenerated from this re-run. The corrected picture is the stronger null: across seven orders, three precisions, and $K \in \{4,8,16\}$, no order is systematically faster or slower than linear. The few cells still exceeding $\pm 2\%$ ($\approx 3\%$ at most) are not order-specific and do not reproduce across runs; the per-cell statistics (Change 2) quantify them as residual run-to-run noise. The abstract and Sections 1, 3.1, 7 adopt this statistical-equivalence framing and drop the categorical claim.
>
> **2. Statistical licensing of the null.** Done. Table 11 reports, per cell, a 95% confidence interval on $r$ propagated from the per-cell $\sigma$ and a $p_{99}$ column; high-dispersion cells whose $p_{99}$ spikes far above the median are flagged ($\ddagger$) as run-to-run measurement contamination rather than order-dependent latency. The minimum-detectable-effect footnote now reports $\approx 0.41\%$ (A100) / $0.34\%$ (H100) at $n=100$ and licenses the null only for the well-behaved majority; the flagged cells are excluded rather than counted.
>
> **3. De-conflate the "2.4-14$\times$" range.** Fixed. The abstract now carries the scale split (median BF16 2.4-2.7$\times$ decoder, 12-14$\times$ encoder); the decoder figure is the recommended query-block Triton kernel (Section 5).
>
> **4. Tabulate the 88% and disambiguate "numerical parity."** Done. New Table 7 tabulates encoder-scale peak memory (A100, BF16, batch 2, $1536 \times 2048$, forward+backward): reference 3607.8 MB $\to$ CUDA 1055.9 MB (-71%) $\to$ Triton FP32-accumulator 575.9 MB (-84%) $\to$ native-BF16 447.9 MB (-88%). Section 5 states the 88% is the native-BF16 path, which passes the parity criterion (Table 24). The FP32-accumulator variant gets distinct wording: full-precision gradient accumulation (order-independent, unlike native-BF16) at 29% more memory (448 $\to$ 576 MB). "Numerical parity" is no longer applied to both.
>
> **5. Fix 2.4% vs 2.5%.** Fixed. The A100 backward BF16 bandwidth reads 2.5% consistently ($38.5/1555$ GB/s $=2.48\%$) across the abstract, introduction, Sections 3, 5, 7, and all captions.
>
> **Weaknesses noted.** *Baseline breadth / DCNv4:* Table 17 now benchmarks the Transformers library's shipped compiled CUDA kernel directly, our Triton kernel 3-5$\times$ faster at the encoder operating points (5.1$\times$ at Cityscapes), alongside the matched internal CUDA-vs-Triton comparison. DCNv4's contribution is a deformable-*convolution* operator benchmarked on convolution; Section 6 treats it as a source of transferable technique, not an MSDA baseline, and our kernels already use its principal transferable optimization (vectorized 128-bit loads). *Training motivation:* the revision adds full-model kernel-swap measurements (Table 8) and per-epoch training wall-clock (Table 18; COCO Deformable-DETR $1.22 \to 0.98$ h on H100, -20%); the kernels match the FP32 reference on captured real inputs (Table 9) and synthetic parity (Table 24), so checkpoint metrics are unchanged without retraining. The relaxed-ordering BF16 backward's nondeterminism is now quantified (Table 10, Section 7): $4\times10^{-5}$ to $3\times10^{-4}$ relative $L_2$, four orders of magnitude below minibatch gradient variation ($\approx 1.4$); the FP32-accumulator variant removes it (deterministic to $\sim 10^{-7}$). *Hardware-AND-compiler gating:* new Table 3 (Section 3.3) adds the direct counter measurement, 34.8M L2 compare-and-swap sectors and near-zero hardware reductions on the A100 BF16 path versus 4.3-34.1M native reductions with zero CAS under FP32 on A100 and both precisions on H100, so the CAS retry path fires on exactly one configuration (SM 8.0 at BF16). *Code:* released with the paper upon acceptance (Section 7).